# A Rheostat of Ceramide and Sphingosine-1-Phosphate as a Determinant of Oxidative Stress-Mediated Kidney Injury

**DOI:** 10.3390/ijms23074010

**Published:** 2022-04-04

**Authors:** Norishi Ueda

**Affiliations:** Department of Pediatrics, Public Central Hospital of Matto Ishikawa, 3-8 Kuramitsu, Hakusan 924-8588, Japan; nueda@mattohp.com; Tel.: +81-76-275-2222

**Keywords:** apoptosis, Bcl-2 family proteins, ceramide, fibrosis, inflammation, kidney injury, mitochondria, proliferation, reactive oxygen species, sphingosine-1-phosphate

## Abstract

Reactive oxygen species (ROS) modulate sphingolipid metabolism, including enzymes that generate ceramide and sphingosine-1-phosphate (S1P), and a ROS-antioxidant rheostat determines the metabolism of ceramide-S1P. ROS induce ceramide production by activating ceramide-producing enzymes, leading to apoptosis, while they inhibit S1P production, which promotes survival by suppressing sphingosine kinases (SphKs). A ceramide-S1P rheostat regulates ROS-induced mitochondrial dysfunction, apoptotic/anti-apoptotic Bcl-2 family proteins and signaling pathways, leading to apoptosis, survival, cell proliferation, inflammation and fibrosis in the kidney. Ceramide inhibits the mitochondrial respiration chain and induces ceramide channel formation and the closure of voltage-dependent anion channels, leading to mitochondrial dysfunction, altered Bcl-2 family protein expression, ROS generation and disturbed calcium homeostasis. This activates ceramide-induced signaling pathways, leading to apoptosis. These events are mitigated by S1P/S1P receptors (S1PRs) that restore mitochondrial function and activate signaling pathways. SphK1 promotes survival and cell proliferation and inhibits inflammation, while SphK2 has the opposite effect. However, both SphK1 and SphK2 promote fibrosis. Thus, a ceramide-SphKs/S1P rheostat modulates oxidant-induced kidney injury by affecting mitochondrial function, ROS production, Bcl-2 family proteins, calcium homeostasis and their downstream signaling pathways. This review will summarize the current evidence for a role of interaction between ROS-antioxidants and ceramide-SphKs/S1P and of a ceramide-SphKs/S1P rheostat in the regulation of oxidative stress-mediated kidney diseases.

## 1. Introduction

Reactive oxygen species (ROS) play a role in normal cellular physiology, including defense against infection, gluconeogenesis, glucose transport, tubuloglomerular feedback, hemodynamics and electrolyte transport [1]. However, in response to diverse stimuli when excess ROS are produced and overwhelm antioxidant redox system, they cause oxidative stress-mediated kidney disease [1,2]. ROS induce cell death such as apoptosis and necrosis and regulate autophagy in the kidney, whereas the antioxidant defense system including antioxidant enzymes and non-enzymatic antioxidants prevents oxidant-induced cell injury [2]. Thus, a rheostat of ROS and the antioxidant defense system is a determinant of oxidant-induced kidney injury.

Sphingolipids (SLs) play a crucial role in the regulation of oxidative stress-mediated disorders [3], including kidney disease [4,5,6]. SLs regulate ROS production and vice versa [6]. ROS and SLs share targeted cellular compartments such as mitochondria [4,5] and cell signaling pathways which regulate oxidant-induced kidney injury [4,5,6]. Thus, a crosstalk between ROS and SLs regulates kidney injury and endothelial dysfunction associated with ROS-mediated chronic kidney disease (CKD) [4,5,6]. Ceramide, the centerpiece of SL metabolism, induces cell death, whereas sphingosine-1-phosphate (S1P) produced by sphingosine kinases (SphKs) has the opposite effect, promoting cell survival [3,4,5]. Thus, a ceramide-S1P rheostat has recently emerged as a determining factor of cell injury in oxidative stress-mediated disorders [3].

Accumulating evidence suggests a role of interaction between ROS-antioxidant and ceramide-S1P and a role of a ceramide-S1P rheostat in the regulation of oxidant-induced kidney injury. This review will summarize the current data supporting a role of interaction between ROS-antioxidants and ceramide-SphKs/S1P and a role of a ceramide-S1P rheostat in the regulation of oxidant-induced kidney injury. This review will focus on various types of oxidative stress-mediated kidney diseases, in which increased ROS production and/or decreased antioxidant levels have been reported elsewhere. For the details regarding the oxidant status in oxidative stress-mediated kidney diseases described here, please refer to previous studies reported in the literature [1,2,6,7,8].

## 2. SLs Metabolism

Ceramide is generated by three major pathways: (1) de novo synthesis, (2) hydrolysis of sphingomyelin (SM) by sphingomyelinase (SMase) and (3) the salvage pathway [3,4,5,6,9] (Figure 1). Ceramide is de novo synthesized at the surface of the endoplasmic reticulum (ER) starting with the condensation of serine and palmitoyl-CoA mediated by serine palmitoyltransferase (SPT), forming 3-ketodihydrosphingosine (3-KdhSph). 3-KdhSph is reduced by 3-KdhSph reductase to dihydrosphingosine (dhSph). DhSph is *N*-acylated by ceramide synthases (CerS1-6), generating dihydroceramide (dhCer), which is converted to ceramide by dihydroceramide desaturase (DES). Once generated, ceramide is transported to the Golgi through the vesicular process or non-vesicular transport by ceramide transfer protein (CERT). Ceramide serves as a substrate for SM production by SM synthase (SMS) and complex glycosphingolipids (GSLs) by GSL synthase. Ceramide is transferred into the Golgi and phosphorylated to ceramide-1-phosphate (C1P) by ceramide kinase (CerK) [9].

SM and GSLs in the Golgi are transported to the plasma membrane by vesicular transport [9]. C1P in the Golgi is delivered to the plasma membrane via C1P transfer protein (C1PTP) [9]. In the plasma membrane, SM is hydrolyzed to yield ceramide by secretory sphingomyelinase (sSMase) and neutral SMase (nSMase) [4,9]. Ceramide is further metabolized into C1P and S1P and converted back into SM by SMS2. SM and GSL residing in the plasma membrane enter the salvage pathway in the acidic compartment of lysosome/the late endosome for the recycling of ceramide [3,4,5,6,9]. In the lysosome/endosome, acid SMase (aSMase) and glycosidase (GCase) produce ceramide from SM and glucosylceramide (GlcCer), respectively, which is hydrolyzed into sphingosine (Sph). Once released into the cytosol, sphingosine re-enters the ceramide synthesis pathway or is phosphorylated into S1P by SphK1/2 followed by breakdown into ethanolamine-1-phosphate (EA1P) and hexadecenal (HD) by S1P lyase (S1PL).

Recently, structure-based molecules of enzymes involved in SLs metabolism including SPT [10] and alkaline CDase (alCDase) [11] have been shown to control the activity and nature of their enzymatic function. In addition, the structural molecules of adiponectin receptors (AdipoR) have intrinsic CDase activity. The binding of adiponectin, an antioxidant, to AdipoR further enhanced its CDase activity [12]. This raises the question as to whether ROS/antioxidants affect the structure of the enzymes involved in SLs metabolism. Further studies are needed to clarify the role of structure-based molecules in the regulation of enzymes involved in SLs metabolism and its modification by ROS/antioxidants.

## 3. The Enzymes That Generate Ceramide and S1P in the Kidney

Table 1 summarizes the localization of enzymes that regulate ceramide formation and SphKs in the kidney [13,14,15,16,17,18,19,20,21,22,23,24,25,26,27,28,29,30,31,32,33,34,35,36,37,38,39,40,41,42,43,44,45,46,47,48,49,50,51,52,53,54,55]. Regarding subcellular localization of enzymes, sphingomyelin synthase (SMS)1/2 is predominantly in the Golgi and to a small extent in the plasma membrane [20]. NSMase exists in the microsomes, Golgi and ER of the kidney [33]. ASMase [28] and acid ceramidase (aCDase) [5,7] are present in the lysosomes. Neutral ceramidase (nCDase) is localized in the plasma membrane, microsomes, Golgi and ER of renal tubular cells (RTCs) [45]. CerS [56], nCDase [57] and mitochondria-associated nSMase (MA-nSMase) [57,58] exist in the mitochondria. ER contains CerS [55,56], DES [9] and SPT [59]. Ceramide produced in the ER can be transported into the mitochondria through mitochondria-associated membranes (MAM) at the mitochondria outer membrane (MOM) [55,56] which are associated with the ER. This process results in a rise in mitochondrial ceramide levels [58]. In addition, ceramide can be transported from the ER to the mitochondria by CERT [59,60].

SphKs are more abundant in the kidney than other organs [61]. SphK1 is predominantly localized in the cytosol [49], microsomes, plasma membranes and nuclei of kidney [49], the lysosomes of podocytes [52], and the cytosol, plasma membrane, mitochondria, lysosomes/peroxisomes, Golgi and ER of RTCs [55]. SphK2 is predominantly localized in the cytosol of kidney, mesangial cells (MCs) and RTCs [50,53,55] and nuclei of MCs [53].

## 4. Interaction between ROS, Antioxidants and Ceramide in Oxidant-Induced Kidney Injury

### 4.1. Increased Ceramide Generation in Oxidant-Induced Kidney Injury

Ceramide contents and activity of enzymes involved in ceramide generation are altered in various types of oxidative stress-mediated kidney diseases (Table 2) [13,22,26,27,31,35,43,44,47,62,63,64,65,66,67,68,69,70,71,72,73,74,75,76,77,78,79,80,81,82,83,84,85,86,87,88,89,90,91,92,93,94,95,96,97,98,99,100,101,102]. Ceramide levels were increased in the kidney tissue of toxic nephropathy [31,62,63,64,65,66,67], nephropathy caused by ultraviolet (UV) irradiation and radiation [13,27,35,65,68,69,70], radiocontrast-induced nephropathy [71], oxalate nephrolithiasis [72,73] and hyperhomocysteinemia [26,74,75]. In myohemoglobinuria and ischemia/reperfusion (I/R) kidney injury [22], levels of ceramide in the kidney were increased, likely due to CerS activation, since activity of aSMase and nSMase was reduced. Similarly, hypoxia/reoxygenation (H/R)-induced ROS activated CerS in RTCs [76,77,78] and SMase in ECs [79]. Ceramide levels were increased via the activation of aSMase in the kidney of anti-glomerular basement membrane (GBM) antibody (Ab)-induced glomerulonephritis (GN) [22] and unilateral ureteral obstruction (UUO) [80,81], although it was challenged [82]. In addition, a CerS inhibitor, fumonisin B1, prevented apoptosis in UUO kidney [71], indicating a role of CerS for increased ceramide generation in UUO. Thus, ceramide contents are increased in oxidant-induced kidney injury, whereas the activation of ceramide-producing enzymes depends on the types of oxidant-induced injury and kidney cells even in response to the same oxidant stimuli.

Knockout of podocyte-specific aCDase gene, Asah1, in mice resulted in an increase in total and C16 ceramide levels in glomeruli, the main substrate of aCDase, leading to foot process effacement (FPE) in podocytes, massive proteinuria and albuminuria, suggestive of nephrotic syndrome (NS) [44]. This indicates that maintenance of ceramide levels in podocyte may play a pathogenic role in the development of NS. In support of this, clinical studies showed that some patients with mutations in SGPL1 encoding S1PL localized in podocytes and MCs developed steroid-resistant NS with increased ceramide levels in serum [83] and fibroblasts via CerS2 activation [84]. Thus, ceramide, which can be controlled by multiple enzymes, plays a crucial role for the maintenance of podocyte function in NS.

Despite the stimulation and inhibition of nCDase by low and high concentrations of advanced glycation endproducts (AGEs), respectively, the levels of ceramide remained unchanged in MCs [47]. However, the levels of ceramide were increased in diabetic kidney via reduced activity of aCDase [43], RTCs [85] and podocytes via SPT [86] and ECs via aSMase [87,88] and CerS [89]. These data suggest that hyperglycemia/AGEs-induced activation of ceramide-producing enzymes may differ among kidney cell types in diabetic nephropathy (DN).

Regarding the change in plasma levels of ceramide, plasma ceramide levels were increased in rats exposed to carbon tetrachloride (CCl4) [28]. Plasma/serum levels of ceramide were also increased in children [90] and adults with chronic GN (CGN) [91]. Similarly, more advanced stages of CKD accompanied by severe albuminuria were associated with increased plasma levels of ceramides [91]. In addition, serum levels of ceramide were higher in patients with lupus nephritis (LN) compared to those with systemic lupus erythematosus (SLE) but no renal injury [92,93], although it was challenged [94]. A rise in serum levels of ceramide due to CerS5 activation in SLE patients, positively correlated with the disease activity and normalized after immunosuppressive treatment [93]. Furthermore, serum protein modifications indicative of total superoxide production were higher in LN patients compared to SLE patients without renal injury [95]. These lines of evidence suggest that the levels of ceramide in circulation could be a biomarker for the disease activity and oxidant-induced kidney injury.

In support of this, plasma levels of fatty acid-esterified ceramides were elevated in diabetic rats, while insulin treatment reduced them [96]. This indicates a pathogenic role of ceramide in the development of DN. Short- and medium-chain ceramides were elevated in the kidney of diabetic mice [97]. Despite increased plasma levels of long-chain ceramides, long-chain and very-long-chain ceramides were decreased in the kidney tissue due to the reduced activity of DES and nSMase in diabetic mice [98]. In addition, albuminuria and renal histological damage were inversely correlated with renal ceramide levels, whereas renal damage was positively correlated with plasma levels of long-chain ceramides [98]. Furthermore, plasma levels of ceramide were higher in diabetic patients with DN than those without and those of long-chain ceramides were correlated with macroalbuminuria [99]. These data suggest that as kidney disease advances, renal synthesis of ceramide may be decreased, while an increase in plasma levels of ceramide may aggregate oxidant-induced kidney injury [100]. However, recent studies showed that despite no difference in plasma levels of long-chain ceramides, those of very-long-chain ceramides were lower in diabetic patients with macroalbuminuria than those without [101]. In addition, urine ceramide levels in diabetic children with DN were higher than those without [102], especially in those with stage 3 DN, and positively correlated with albuminuria [103]. This suggests that plasma/serum levels of ceramide can be affected by its excretion into urine. Thus, a decrease in renal ceramide synthesis and an alteration of plasma/serum levels of ceramide, which are modulated by urine excretion of ceramide, can be a biomarker of severity of the diseases. Further studies are needed to address this issue.

### 4.2. ROS Activate and Translocate the Enzymes That Generate Ceramide

ROS can regulate ceramide-producing enzymes in the kidney (Figure 2). Superoxide, hydrogen peroxide (H_2_O_2_) and nitric oxide (NO) increased ceramide production in MCs [24,104] and GECs [104]. Hydrogen peroxide activated SMase [72] and CerS [77] in RTCs. Hydrogen peroxide directly phosphorylated the serine in position 173 of nSMase, leading to its activation in other types of cells [105]. Similarly, nitric oxide activated aSMase and nSMase in MCs [24] but inhibited aCDase and nCDase [24]. However, it remains to be determined whether and how ROS directly activate ceramide-producing enzymes or inhibit ceramide-degrading enzymes.

ROS can induce ceramide formation through the induction of subcellular translocation of ceramide-producing enzymes. Superoxide induced the lysosome trafficking of aSMase by lysosome exocytosis, leading to ceramide release and formation of ceramide-enriched membrane platforms [106]. This formation of lipid rafts (LRs), where superoxide can activate aSMase, further promoted LR clustering, resulting in intrinsic signaling amplification of the platforms in ECs [106]. In addition, high glucose facilitated the lysosome–membrane fusion, LR clustering and LR-NADPH oxidase (NOX) subunit platforms in ECs, where superoxide and ceramide can be produced [107]. However, these events were inhibited by the inhibition of lysosome-membrane fusion or by silencing aSMase located at the lysosome. These data indicate that hyperglycemia-induced EC dysfunction is related to the lysosome-membrane fusion and subsequent LR clustering, LR-NOX platforms formation and O_2_^−^ production, which in turn activates aSMase, resulting in ceramide formation. Furthermore, TNF-α-related apoptosis inducing ligand (TRAIL) induced translocation of aSMase, ceramide generation and NADPH oxidase activation in the membrane raft clusters followed by lysosomal fusion with the plasma membrane in an aSMase-dependent manner, which resulted in EC dysfunction, while these events were abolished in aSMase-knockout ECs [108]. Thus, ROS-induced translocation of ceramide-producing enzymes may affect ceramide-induced cell death. In support of this, cisplatin, an oxidative stress, induced CerS1 translocation from the ER to the Golgi in RTCs, which resulted in increased susceptibility to cisplatin toxicity [109]. These data suggest that ROS induce the translocation of ceramide-producing enzymes, which in turn regulates ceramide-induced cell death in oxidant-induced kidney injury (Figure 2). ROS can activate ceramide-producing enzymes through ROS-induced downstream signaling pathways. Nitric oxide degraded nCDase via the inhibition of PKCδ and PKCα in MCs [25]. Oxidative stress produced arachidonic acid (AA) via the activation of phospholipase A2 (PLA2), which inhibits CDase activity and stimulates SMase activity, leading to ceramide production in RTCs [110]. In addition, oxidized low-density lipoprotein (oxLDL) activated nSMase-2 in ECs, and this process required lectin-like oxLDL receptor-1 (LOX-1), ROS production by nicotinamide adenine dinucleotide phosphate (NADPH) oxidase and activation of p38 mitogen-activated protein kinase (p38MAPK) [34]. Furthermore, p53 stimulated nSMase activity through superoxide formation in other types of cells [111]. Thus, ROS-induced signaling pathways play a crucial role in the regulation of activity of ceramide-producing enzymes. ROS-induced proinflammatory cytokines can activate ceramide-producing enzymes. ROS induce the production of proinflammatory cytokines such as tumor necrosis factor (TNF)-α and interleukin (IL)-1β [112] and vice versa [113] (Figure 2). Once generated following oxidant stimuli, TNF-α activated SMase in MCs [48] and CerS in RTCs [114]. IL-1β increased ceramide formation in MCs by activating SMase [115]. Conversely, ceramide stimulated production of TNF-α and IL-1β [116], which further enhances ROS production and activity of ceramide-producing enzymes. These data suggest that ROS can activate ceramide-producing enzymes and inhibit ceramide-degrading enzymes directly or through ROS-induced signaling pathways.

### 4.3. Antioxidants Regulate the Enzymes That Generate Ceramide

Antioxidants can regulate ceramide-producing enzymes. Decreased glutathione (GSH) and GSH/glutathione disulfide (GSSG) ratio enhanced nSMase activity [31,33]. Oxalate-induced ceramide production accompanied by decreased SM contents was blocked by an antioxidant N-acetylcysteine (NAC) and a superoxide dismutase (SOD) mimetic but not by a CerS inhibitor fumonisin B1 [72], suggesting that antioxidants may suppress the activity of SMase but not CerS. H_2_O_2_ induced the depletion of GSH and glutathione S-transferase (GST), which in turn enhanced the activity of SMase and SPT in RTCs, while GSH and NAC ameliorated these events [117]. Similarly, antioxidants inhibited superoxide-induced activation of aSMase [106] and GSH inhibited nSMase activity [33]. In addition, GSH can directly inhibit nSMase in RTCs [31,33,118] and ECs [119], while reduced GSH can inhibit CDase [120]. These data suggest that a rheostat of ROS-antioxidants can regulate activity of ceramide-producing enzymes (Figure 2).

Nuclear factor-erythroid 2-related factor-2 (Nfr2) is the most important inducible transcription factor that exerts protective effects against oxidant-induced kidney injury by stimulating the endogenous antioxidants. Once activated in cytosol, Nrf2 transactivated antioxidant response elements, which further enhanced the expression and activity of antioxidants such as SOD, GSH, glutathione peroxidase (GPx), GST, catalase and heme oxygenase (HO)-1 [121]. I/R decreased the expression of Nrf2 and HO-1 in the kidney, leading to apoptosis/necrosis. Meldonium, an anti-ischemic drug clinically used to treat myocardial and cerebral ischemia that shifts energy production from fatty acid oxidation to less oxygen-consuming glycolysis, increased the expression of Nrf2 and HO-1, thereby increasing the expression and activity of antioxidants such as SOD, GPx and GST, leading to protection against I/R-induced kidney injury [121]. Since meldonium inhibited I/R-induced renal formation of fatty acids [121], which are a basic component of all lipids including palmitate and SLs (e.g., ceramide), it would be interesting to determine whether this agent suppresses ceramide levels by inhibiting supply of fatty acids which is needed for ceramide synthesis [122]. In addition, it would be interesting to find out whether meldonium inhibits ceramide synthesis via modulation of the enzymes involved in ceramide metabolism induced by antioxidants through Nrf2 and HO-1 in oxidant-induced kidney injury. Further studies are needed to determine whether antioxidants and antioxidant agents that activate Nfr2 modulate enzymes involved in ceramide metabolism directly or through the inhibition of ROS-induced signaling pathways.

### 4.4. Ceramide Stimulates ROS Production and Inhibits Antioxidant Defense System Which Further Activates Ceramide-Producing Enzymes

Following exposure to oxidant stimuli, accumulated ceramide including a rise in mitochondrial ceramide resulted in mitochondrial dysfunction, leading to mitochondrial ROS production and reduced antioxidant thiol proteins in RTCs [62,86]. Ceramide formation via aSMase [26,70] and SPT [74] induced ROS production by activating Rac guanosine triphosphatase (GTPase) and NADH/NADPH oxidase in oxidant-induced kidney injury, while an inhibitor of SPT, myriocin, which inhibits de novo ceramide synthesis, and an inhibitor of NADPH oxidase, apocynin, inhibited ROS generation and glomerular injury [74,75]. These data suggest that ceramide can activate Rac GTPase/NADPH oxidase, leading to mitochondrial dysfunction and ROS production in oxidant-induced kidney injury.

In support of this, activation of aSMase not only increased ceramide production but also activated NADPH oxidase activity, leading to ROS production, while these were blocked by an aSMase inhibitor, amitriptyline, in GECs [28]. Ceramide by itself enhanced ROS production in ECs [79,123]. Activation of nSMase induced the translocation of endothelial nitric oxide synthase (eNOS) from the plasma membrane to intracellular region, eNOS phosphorylation and ceramide formation in ECs [124]. In addition, TNF-α-induced ROS production was mediated by ceramide formation due to aSMase activation in ECs [125], whereas an antioxidant, NAC, inhibited ROS production caused by SMase-induced ceramide formation in MCs exposed to TNF-α [113,126]. Furthermore, exogenous SMase administration stimulated ROS production in MCs, while the antioxidant NAC inhibited these events [126]. Thus, following exposure to oxidant stimuli, accumulated ceramide caused by ROS-induced activation of ceramide-producing enzymes can activate Rac GTPase/NADPH oxidase, resulting in further enhancement of ROS production, whereas antioxidants can inhibit these events.

On the other hand, ceramide inhibited the redox system, including Nfr2, which preserves the redox balance and controls the expression of antioxidant enzymes such as GST [127]. This implies that during oxidative stress, ceramide suppresses the antioxidant defense system. In support of this, the levels of ceramide were increased, but the expression of Nfr2 and antioxidants such as GPx and catalase was reduced, thereby aggravating cisplatin nephrotoxicity [128]. In contrast, treatment with Nrf2 inducer, methyl-2-cyano-3,12- dioxooleano-1,9-dien-28-oate (CDDO), at the initial phase of I/R ameliorated RTC injury by inducing the antioxidant enzymes [129]. Thus, ceramide not only stimulates ROS production but inhibits antioxidants in oxidant-induced kidney injury (Figure 2). These lines of evidence suggest that a crosstalk between ROS/antioxidants and ceramide-producing enzyme activity/ceramide formation regulates oxidant-induced kidney injury.

## 5. Role of Mitochondrial Function in the Regulation of Ceramide and ROS Formation

### 5.1. Interaction between Bcl-2 Family Proteins and the Enzymes That Generate Ceramide

Bcl-2 family proteins, the major regulators of the mitochondrial pathway for apoptosis, are mainly localized in the mitochondria but also in other intracellular compartments such as the ER, Golgi, nucleus and peroxisomes [130]. There is a crosstalk between ROS and Bcl-2 family proteins during oxidative stress [131] (Figure 2). Oxidant stimuli including H_2_O_2_ stimulate the expression of apoptotic (Bax, Bad, Bcl-Xs and Bim) but inhibit anti-apoptotic Bcl-2 family proteins (Bcl-2 and Bcl-xL) [132,133]. Pro- and anti-apoptotic Bcl-2 family proteins can activate or inhibit the activity of ceramide-producing enzymes, respectively. Pro-apoptotic Bak but not Bax was required for CerS activation in the mitochondria [65,134] and stimulated ceramide generation through post-translational regulation of specific CerS isoforms [65]. Bak together with truncated Bid (t-Bid) more potently activated CerS activity, leading to a further increase in ceramide levels, ceramide channel formation in the mitochondria and inhibited Bcl-2 expression, resulting in mitochondrial dysfunction [134]. Bax/Bak also stimulated mitochondrial ceramide formation in baby mouse kidney (BMK) cells [135], while knockout of Bax and Bak inhibited UV-induced activation of CerS5 and CerS6 in other types of cells [136]. These data suggest that apoptotic Bax/Bak/Bid can activate CerS isoforms. Conversely, UV light-induced activation of aSMase was required for Bax conformational change at the mitochondrial membrane, leading to mitochondrial release of apoptotic factors [137].

In contrast, anti-apoptotic Bcl-2 proteins can inhibit ceramide-producing enzymes. Overexpression of Bcl-xL inhibited activity of CerS5 and CerS6 [136], suggesting that the inhibitory action of Bcl-xL has selectivity of CerSs. Similarly, overexpression of Bcl-2 and Bcl-xL inhibited ceramide formation through the inhibition of nSMase in non-stimulated cells or cells exposed to TNF-α and cisplatin [138]. In addition, inhibition of ceramide production may require a high level of Bcl-2 expression which acts as upstream and downstream of ceramide [138]. Taken together, these data suggest a crosstalk between apoptotic/anti-apoptotic Bcl-2 proteins and ceramide-producing enzymes in oxidant-induced kidney injury.

Since apoptotic Bcl-2 family proteins can activate ceramide-producing enzymes, while anti-apoptotic Bcl-2 proteins have the opposite effect, interaction and balance between apoptotic and anti-apoptotic Bcl-2 proteins can regulate the activity of ceramide-producing enzymes. Inhibition of anti-apoptotic Bcl-2 proteins (Bcl-2, Bcl-xL and Bcl-w) activated Bak and Bax, leading to the activation of CerS in the mitochondria [134]. In addition, nSMase cooperated with pro-apoptotic C8-Bid to promote mitochondrial outer membrane permeability (MOMP), which was blocked by BcL-xL, and MOMP was promoted via nSMase activation through Bak [139]. Thus, the interaction between ceramide-producing enzymes and pro and anti-apoptotic Bcl-2 family proteins not only regulates ceramide formation but also mitochondrial function, thereby affecting the process of apoptosis in oxidant-induced kidney injury. Further studies are needed to determine the mechanism by which Bcl-2 family proteins regulate ceramide-producing enzymes in oxidant-induced kidney injury.

### 5.2. Bcl-2 Family Proteins Regulate Ceramide-Induced ROS Production

Bcl-2 family proteins can regulate ceramide-induced ROS production and vice versa (Figure 2 and Figure 3). Anti-apoptotic Bcl-2 functions as an antioxidant [140] and thus inhibits ROS-induced ceramide formation. In support of this, in ECs exposed to H/R, ceramide-induced ROS production via aSMase activation was inhibited by Bcl-2 due to the prevention of superoxide production rather than a direct electron-scavenging or superoxide-metabolizing activity of Bcl-2 itself [79]. Overexpression of the BCL-2 gene also inhibited GSH oxidation and ceramide formation induced by irradiation in other types of cells [141].

Conversely, antioxidants inhibit apoptotic Bcl-2 proteins and stimulate anti-apoptotic Bcl-2 proteins, thereby inhibiting ROS-induced ceramide formation. This has been supported by the following observations. First, overexpression of antioxidants such as glutathione peroxidase 1 (GPX1) in ECs [142], HO-1 [143] and SOD [144] reduced the Bax/Bcl-2 ratio in I/R injury of other types of cells. In addition, meldonium, which functions as an antioxidant, decreased the ratio of Bax/Bcl-2, ameliorating I/R-induced kidney injury [121]. An antioxidant, rutin, inhibited CCl4-induced ceramide and ROS production and increased Bcl-2 expression in the kidney [63]. Furthermore, α-mangostin, a xanthone natural product, and desipramine, an inhibitor of SMase, that function as antioxidants prevented high glucose-induced Bax upregulation, Bcl-2 downregulation, increased ceramide levels via aSMase activation, ROS production and apoptosis in ECs [88]. Finally, exogenous C2-ceramide induced ROS production and decreased GSH and Bcl-2 expression, leading to caspase-3 activation and apoptosis in ECs, while antioxidants including GSH prevented these events [123]. Thus, antioxidants can prevent ROS-induced ceramide production by upregulating anti-apoptotic and downregulating apoptotic Bcl-2 family proteins. Further studies are needed to determine the mechanism by which Bcl-2 proteins prevent ROS-induced ceramide formation in oxidant-induced kidney injury.

## 6. Role of Mitochondria and Cell Signaling Pathways for Ceramide-Induced Apoptosis in Oxidant-Induced Kidney Injury

### 6.1. Ceramide-Induced Cell Death in Oxidant-Induced Kidney Injury

Ceramide can induce apoptosis and/or necrosis in various types of kidney cells including ECs associated with oxidative stress-mediated kidney diseases (Table 2) [13,22,26,27,31,35,43,62,63,65,66,67,68,69,70,71,72,73,74,75,76,77,78,79,85,86,87,88,89,90,91,92,93,94,97,98,99,100,101,102,103]. In support of this, exposure of MCs, GECs [104] and RTCs [77] to ROS [104] increased ceramide production, leading to apoptosis/necrosis, while antioxidants prevented ROS-induced ceramide production and apoptosis in ECs [70]. Thus, a crosstalk between ceramide and redox signaling contributes to the progression of oxidant-induced kidney injury.

Increased ceramide levels in podocytes from aCDase-knockout mice [44] and in steroid-resistant NS patients with mutations of SGPL1 [83,84] led to foot process effacement (FPE) indicative of NS and focal segmental glomerulosclerosis (FSGS). Similarly, increased ceramide levels were associated with apoptosis [43,85,86] and necrosis [99,101,102,103] in the kidney of DN. Podocytes under high glucose increased mitochondrial ceramide and ROS production, while a selective SPT inhibitor, myriocin, prevented these events and disruption of mitochondrial integrity, podocyte apoptosis and glomerular injury in DN [86]. In addition, plasma levels of the long-chain ceramides (C16:0, C18:0, C20:0) were positively correlated with albuminuria and renal histology in DN [98]. Furthermore, increased levels of ceramide caused EC dysfunction, while inhibition of CerS restored EC function and atherosclerosis in DN [89]. These data suggest that ceramide regulates maintenance of the structural and functional integrity of the kidney cells in oxidant-induced kidney injury.

Clinical studies show that ceramide levels in plasma and kidney are correlated with severity of disease activity and can be a predicting factor for the progression of the disease. For example, ceramide induced necrosis in the kidney of CGN [90,91], and higher plasma levels of ceramides were associated with more advanced stages of CKD accompanied by abnormal albuminuria [91]. Similarly, in LN patients with renal dysfunction and proteinuria, plasma levels of ceramide were positively correlated with proteinuria and kidney disease activity [92,93]. These data suggest that plasma levels of ceramide can be a predicting biomarker of the disease activity.

### 6.2. A Role of Mitochondria for the Regulation of ROS/Antioxidants and Ceramide Generation

Mitochondria play a crucial role in oxidant-induced kidney injury (Figure 3). Oxidant stress inhibits mitochondrial respiratory chain complex (MRCC) which induces ROS production [145]. Conversely, ROS inhibits MRCC [146], while inhibition of MRCC increases ceramide formation [147]. In addition, accumulated ROS production following oxidant stimuli resulted in increased ceramide levels [62], loss of mitochondrial membrane potential (MMP), mitochondrial permeability transition pore (MPTP) opening, MOMP, decreased expression of voltage-dependent anion channel (VDAC), suggestive of its closure, decreased Bcl-2 and increased Bax expression, leading to mitochondrial cytochrome C (Cyto C) release and apoptosis [148]. Hydrogen peroxide induced increased Bax and decreased Bcl-2 expression and a rise in mitochondrial Ca^2+^ by enhancing the transfer of Ca^2+^ from the ER, leading to increased mitochondrial Ca^2+^ and subsequent apoptosis in RTCs [149]. Hydrogen peroxide also increased intracellular Ca^2+^ [150,151], which further enhanced ROS production [150], while the overexpression of Bcl-2 inhibited these events and apoptosis by increasing the capacity of mitochondria to store and buffer Ca^2+^ [151]. However, inhibition of MPTP opening and MOMP prevented ROS production and restored the antioxidant level [152], suggesting that mitochondrial function regulates a rheostat of ROS and antioxidants and subsequent cell fate.

In contrast, antioxidants restored MRCC [153], prevented the loss of mitochondrial membrane potential (MMP) [148,154,155] and increased the expression of VDAC [148,154] and Bcl-2/Bax ratio [63,88,148,156], as well as preventing an increase in intracellular Ca^2+^ which induces loss of MMP [155], thereby ameliorating apoptosis in oxidant-induced kidney injury. In fact, antioxidants prevented a cadmium-induced increase in intracellular Ca^2+^ [155] and increased Bcl-2 expression, which prevents a rise in intracellular Ca^2+^ [156], thereby ameliorating apoptosis. These data suggest that mitochondria regulate a rheostat of ROS and antioxidants and vice versa in oxidant-induced kidney injury.

Interaction between ROS/antioxidants and ceramide can regulate mitochondrial function. The inhibition of MRCC induced ROS production [145,157], which further enhanced ceramide generation [147], suggesting that mitochondrial dysfunction induces ROS production, which subsequently stimulates ceramide formation. Ceramide inhibits MRCC, leading to the production of ROS [145]. Podocytes under high glucose increased mitochondrial ceramide and ROS production. A ceramide synthesis inhibitor, myriocin, prevented mitochondrial ROS generation and apoptosis [86], suggesting that mitochondrial ceramide accumulation can further stimulate ROS production, leading to the progression of mitochondrial dysfunction. Both ROS and ceramide activated Bax or Bak, which in turn enhanced mitochondrial uptake of Ca^2+^ through sarcoplasmic-ER Ca^2+^ adenosine triphosphatase (SERCA), and these events triggered MPTP opening, leading to apoptosis in other type of cells [158].

Antioxidants inhibited TNF-α-induced ROS production by inhibiting SMase-induced ceramide production in MCs [126]. Depletion of antioxidant, GSH, prior to exposure of the mitochondria to ceramide enhanced ROS generation, leading to lipid peroxidation and loss of MRCC [153]. These data suggest that antioxidants prevent ceramide-induced mitochondrial ROS production by restoring MRCC and that mitochondrial function regulates a rheostat of ROS and antioxidants and vice versa, which in turn modulates ceramide formation. In addition, an interaction between ROS/antioxidants and ceramide contributes to mitochondrial dysfunction, leading to apoptosis.

### 6.3. Ceramide-Induced Mitochondrial Dysfunction Leading to Apoptosis

Mitochondria play a central role for the regulation of ceramide-induced apoptosis in oxidant-induced kidney injury (Figure 4) [4]. Accumulated mitochondrial ceramide induced mitochondrial dysfunction including inhibition of MRCC and reduced Bcl-2/Bax ratio, resulting in further ROS generation, decreased antioxidants and subsequent apoptosis [63,71,73,79,88,123]. H/R-induced ceramide production via the inhibition of MRCC III preceded ROS production, and was inhibited by a SMase inhibitor, desipramine, a MRCC III inhibitor, antimycin A, and Bcl-2 [79]. In addition, ceramide suppressed MRCC IV, resulting in ROS production [145,153], thereby inducing MOMP and reducing MMP, leading to apoptosis, whereas these events were inhibited by Bcl-2 [79]. Furthermore, ceramide sensitized the MPTP opening to Ca^2+^, releasing Cyto C from the mitochondria in RTCs [159], while the restoration of MRCC II inhibited Ca^2+^-dependent MPTP opening, ameliorating I/R kidney injury [160]. These lines of evidence suggest that ceramide can inhibit MRCC, which results in ROS production and mitochondrial dysfunction, leading to apoptosis, and that these events can be inhibited by anti-apoptotic Bcl-2 proteins.

Ceramide stimulates apoptotic and inhibits anti-apoptotic Bcl-2 family proteins, leading to mitochondrial dysfunction and apoptosis. In support of this, exogenous C2-ceramide increased the expression of Bax accompanied by reduced Bcl-2 expression [71] and mitochondrial translocation of Bax, leading to apoptosis in RTCs [161]. Ceramide aggravated Bax-induced MPTP opening and oligomeric Bax promoted it, leading to the release of Ca^2+^ and Cyto C from the mitochondria in the kidney [162]. In addition, ceramide together with Bax-induced MOMP [163] and opening of MPTP which sensitized the mitochondria to Ca^2+^ [162]. Furthermore, ceramide induced the release of Ca^2+^ from the ER, resulting in increased mitochondrial uptake of Ca^2+^, leading to a change in mitochondrial morphology and apoptosis, while Bcl-2 decreased ceramide-induced Ca^2+^ release from the ER [164]. These data suggest that ceramide can increase the ratio of apoptotic/anti-apoptotic Bcl-2 family proteins which results in mitochondrial dysfunction and mitochondrial Ca^2+^ uptake, leading to apoptosis.

Ceramide can regulate two major forms of channels, namely the ceramide channel and VDAC in the mitochondrial outer membrane (MOP), leading to apoptosis. Ceramide channels are able to pass large proteins to exit mitochondria during apoptosis [165] which trigger MPTP opening and MOMP, leading to Cyto C release, caspase activation and apoptosis [166]. Apoptotic Bcl-2 proteins can promote ceramide channel formation by activating ceramide-producing enzymes. For example, Bak activated CerS, accelerating ceramide channel formation [134,166], while anti-apoptotic Bcl-xL disassembled them [166]. In contrast, anti-apoptotic Bcl-2 proteins, Bcl-xL and CED-9, disassembled ceramide channels by direct interaction, preventing MOM permeabilization (MOMP) and apoptosis [166]. Thus, apoptotic Bcl-2 family proteins stimulate ceramide channel formation, while anti-apoptotic Bcl-2 proteins have the opposite effect. In support of this, location of Bax and high concentrations of ceramide have been shown in the mitochondrial membrane from the ischemic organ, which may facilitate ceramide channel formation [167].

Ceramide regulates the function of VDAC in the MOM which regulate apoptosis. Binding of ceramide to VDAC, which serves as a platform for the mitochondrial recruitment of Bax and Bak, induced MOMP and apoptosis in human embryonic kidney (HEK) cells [168]. Interaction between ceramide, Bax and VDAC in the MOM induced apoptosis-inducing factor (AIF) release from the mitochondria, leading to apoptosis [169]. Ceramide binds to tubulin and this formation leads to closure of VDAC1, thereby reducing mitochondrial ATP release in other types of cells [170]. Furthermore, ceramide dephosphorylated Bad which was activated by protein phosphatase 2A (PP2A) but inhibited by PKA/PKC [159]. Dephosphorylated Bad was required for ceramide-induced sensitization of MPTP opening and more Bad and less VDAC were associated with Bcl-xL at the mitochondrial membrane, independent of Bax/Bak. These events cause sensitization of MPTP to Ca^2+^, leading to the release of Cyto C from the mitochondria and apoptosis [159]. Taken together, these data suggest that mitochondria play a crucial role in the regulation of ceramide-induced apoptosis in oxidant-induced kidney injury.

### 6.4. Ceramide-Induced Cell Signaling Pathway for Cell Death

Ceramide can regulate multiple signaling pathways, leading to apoptosis in oxidant-induced kidney injury (Figure 5). Mitogen activated protein kinases (MAPKs), consisting of extracellular signal-regulated kinase (ERK), p38MAPK and c-Jun N-terminal kinase (JNK), play an important role in the regulation of ceramide-induced apoptosis. ROS-induced ceramide activated ERK, p38MAPK, JNK, proinflammatory cytokines (TNF-α, IL-1β, IL-6), nuclear factor (NF)-κB, p53, calpain, increased Bax/Bcl-2 ratio and inhibition of antioxidant, HO-1, leading to Cyto C release, caspase-3 activation and apoptosis. An antioxidant, rutin, prevented ROS/ceramide production and these signaling events described above and increased the Bcl-2/Bax ratio, ameliorating apoptosis in oxidant-induced kidney injury [63,64]. These factors, except for ERK, function as apoptotic in ceramide-induced apoptosis with oxidant kidney injury, as described below.

ROS-induced ceramide suppressed vascular endothelial growth factor (VEGF) expression, leading to apoptosis [26,75], whereas aSMase-knockout attenuated these events [75]. VEGF prevented ceramide-induced apoptosis by activating ERK [171]. In addition, ceramide activated ERK in ECs [172], whereas ERK by itself functions as anti-apoptotic [172] or antagonistic to p38MAPK activation [109], ameliorating ceramide-induced apoptosis [173]. Thus, ERK functions as anti-apoptotic in oxidant-induced kidney injury. In contrast, ROS-induced ceramide activated p38MAPK, which resulted in the apoptosis of RTCs [108] and ECs [69]. OxLDL mediated by ROS produced ceramide, which in turn activated p38MAPK, leading to apoptosis in vascular smooth muscle cells (VSMC) [173], contributing to atherosclerosis in CKD. In addition, C6-ceramide or bacterial SMase activated p38MAPK, leading to apoptosis in ECs [69]. These data suggest that p38MAPK functions as apoptotic in ceramide-induced apoptosis.

UV- and oxLDL-induced ceramide activated JNK, leading to apoptosis in HEK cells [27] and VSMC [173]. While ceramide failed to activate JNK in MCs [174] and RTCs [161], ceramide activated JNK in GECs, leading to apoptosis [174]. These data suggest that activation of JNK by ceramide depends on kidney cell types. In addition, ceramide activated stress-activated protein kinase (SAPK)/JNK through stimulation of PKCζ, which formed a complex of mitogen-activated protein kinase kinase kinase 1 (MEKK1)/SEK/SAPK, leading to cell growth arrest in HEK cells [175]. However, IGF-I inhibited the interaction between ceramide-induced SAPK/JNK and PKCζ, suggesting that IGF-I can suppress ceramide-activated SAPK/JNK, ameliorating ceramide-induced apoptosis. Taken together, these data suggest that JNK functions as apoptotic in ceramide-induced apoptosis in oxidant-induced kidney injury, while ceramide-induced JNK activation depends on kidney cell types.

The onco-suppressor p53 is a transcription factor that regulates a wide spectrum of genes involved in cellular functions including apoptosis. p53 is the mediator of guanosine triphosphate (GTP) depletion-induced apoptosis in I/R kidney injury [176]. Binding of ceramide to p53 induces its activity, leading to apoptosis [177]. These data suggest that ceramide-induced p53 activation functions as apoptotic in oxidant-induced kidney injury.

Another transcription factor, NF-κB, functions as apoptotic in ceramide induced apoptosis [178]. Cadmium-induced ROS and ceramide increased Ca^2+^-dependent calpain activity through ceramide-induced increase in intracellular Ca^2+^ [62], downregulated antioxidants and Bcl-2 and upregulated Bax, proinflammatory cytokines (TNF-α, IL-6), NF-κB, nitric oxide and oxidative stress, leading to apoptosis [179], while antioxidants blocked these events [179]. Ceramide and hydrogen peroxide activated NF-κB [180] which aggravated cadmium- [179] and hypoxia-induced apoptosis in RTCs [181]. In addition, ceramide potentiated TNF-α-induced NF-κB activation in MCs, leading to apoptosis [126]. These data suggest that NF-κB and Ca^2+^-dependent calpain as well as proinflammatory cytokines function as apoptotic in oxidant-induce kidney injury.

Oxalate induced ceramide production, mitochondrial dysfunction, mitochondrial ROS production and apoptosis in RTCs. These events were blocked by the inhibition of cytosolic PLA2 (cPLA2) [182], suggesting that oxalate-induced PLA2 activation triggers mitochondrial dysfunction, leading to ceramide-induced apoptosis in RTCs. PLA2 cleaves AA and COX-2 converts to AA in the kidney [183]. Oxidative stress enhanced cPLA2 activity and the binding of ceramide to cPLA2 increased AA release [184]. Both cPLA2 and AA increased ceramide production in RTCs, resulting in apoptosis [110,182]. Furthermore, ceramide can directly activate PLA2 [185], and enhance PLA2-induced cytotoxicity in RTCs [186]. In addition, TNF-α-induced ceramide activated soluble PLA2 (sPLA2), cyclooxygenase (COX)-2 [113] and cPLA2 [185] in MCs, leading to apoptosis [126,187] or necrosis [187]. Taken together, these data suggest that ceramide activates PLA2/AA/COX-2 to promote ceramide-induced apoptosis in oxidant-induced kidney injury.

A CerS inhibitor, myriocin, or a NADPH oxidase inhibitor, apocynin, decreased hypermonocysteinemia-induced tissue inhibitor of matrix metalloproteinase-1 (TIMP-1) that decreases MMP-1. This inhibition of TIMP-1 subsequently resulted in increased expression of MMP-1, ameliorating hypermonocysteinemia-induced glomerular injury [74]. These data suggest that ceramide stimulates the expression of TIMP-1, thereby inhibiting MMP-1 and leading to oxidant-mediated glomerular injury.

Protein phosphatase 2A (PP2A), which is a major Ser/Thr phosphatase involved in several cellular signal transduction pathways, functions as apoptotic in ceramide-induced apoptosis. Ischemia-induced and exogenous ceramide suppressed PKC-α, which inhibits PP2A, thereby upregulating PP2A B56a, which resulted in apoptosis/necrosis in NRK-52E cells [188]. Microcystin-LR (MCLR), which induces ROS and ceramide, activated the expression and activity of PP2A in RTCs and the kidney, leading to apoptosis, while exogenous ceramide produced similar findings [189]. Thus, ceramide-induced PP2A promotes apoptosis in oxidant-induced kidney injury.

Serum- and glucocorticoid-inducible protein kinase (SGK)-1 has a variety of cellular functions, including a regulatory role of apoptosis [190]. SGK family members share a similar structure, substrate specificity and function with Akt and signal downstream of the phosphatidylinositol 3-kinase (PI3K) signaling pathway [191]. In addition, ceramide activated SGK-1 without increasing its phosphorylation via p38MAPK/cyclic AMP (cAMP)/protein kinase A (PKA)/phosphoinositide-3-kinase (PI3K) and decreased Akt activity in HEK cells [114]. However, overexpression of anti-apoptotic SGK-1 protected C2-ceramide- and TNF-α-induced apoptosis by activating PI3K [114]. These data suggest that SGK-1 functions as anti-apoptotic by activating PI3K/Akt in ceramide-induced apoptosis. In support of this, overexpression of SGK-1 ameliorated I/R- and H/R-induced kidney injury and RTC apoptosis [192], as well as ROS production and apoptosis in ECs under high glucose [192,193].

A CerS inhibitor, fumonisin B1, reversed radiocontrast-induced ceramide formation which inhibited Akt/cAMP response element-binding protein (CREB) and Bcl-2, leading to apoptosis in RTCs [71], suggesting that Akt/CREB by itself functions as anti-apoptotic in ceramide-induced apoptosis. In addition, ROS-induced IL-1β stimulated ceramide formation via nSMase in MCs [32,115], while these were inhibited by PKC [32]. This implies that PKC inhibits pro-inflammatory cytokines-induced ceramide formation, thereby inhibiting ceramide-induced apoptosis. Furthermore, ceramide enhanced the serine/threonine kinase, LIM kinase-1 (LIMK-1), that regulates cytoskeletal organization in MCs, leading to apoptosis [115]. Thus, PI3K/Akt/CREB/PKC functions as anti-apoptotic, while LIMK-1 has the opposite effect in the regulation of ceramide-induced apoptosis.

Taken together, these data suggest that ROS-induced ceramide activates proinflammatory cytokines, p38MAPK/SAPK/JNK/NF-κB, p53, Ca^2+^-dependent calpain, PLA2/COX-2/AA, TIMP-1, PP2A and LMK-1, leading to apoptosis, while ERK, VEGF, MMP-1, SGK-1 and the PI3K/Akt/CREB/PKC pathway prevent ceramide-induced apoptosis in oxidant-induced kidney injury.

## 7. Interaction between ROS and S1P in Oxidant-Induced Kidney Injury

### 7.1. Alteration of S1P Levels in Oxidative Stress-Mediated Kidney Disease

Table 3 shows the data for S1P levels in oxidative stress-mediated kidney diseases, including unchanged levels in the kidney of cadmium nephrotoxicity [131], decreased levels in podocytes after radiation [35] and increased levels in the kidney of UUO [194], I/R [195,196], RTCs [195] and ECs [197] exposed to H/R. S1P levels were also increased in diabetic kidney [46,198,199,200], glomeruli [46] and MCs [47,199,200], although contradictory results were reported in diabetic kidney [43]. Some steroid-resistant NS patients with mutations in SGPL1 encoding S1PL had below the limits of detection of S1P in fibroblast-conditioned medium [84]. These data suggest that the change in S1P levels depends on the types of kidney cells and oxidant stimuli.

In terms of SphK subtypes, the expression of SphK1 [201,202] and SphK2 [194] was increased in UUO kidney, although it was challenged [81]. However, despite decreased expression of SphK1 in HK-2 cells exposed to H/R [195], the expression of SphK1 but not SphK2 was increased in I/R kidney [196] and ECs [197]. These data suggest that activation of SphK1/2 depends on kidney cell types even in response to the same oxidant stimuli. In addition, increased expression of S1PR1/2/5 was found in the spleen from mice with anti-GBM Ab GN [203]. The expression and activity of SphK1 were increased in diabetic kidney [198,200,204], MCs [198,204], RTCs [94] and VSMC [205], while SphK2 remained unchanged in diabetic kidney, MCs [198] and VSMC [205]. SphK1 was overexpressed in podocyte from diabetic patients, while SphK1-deficient mice developed a more severe DN [206], suggesting a protective role of SphK1 for podocytopathy in DN. However, SphK2-knockdown has recently been shown to protect against diabetic podocyte injury and albuminuria [207]. These data indicate that SphK1 may protect and SphK2 may contribute to the progression of DN.

Regarding the levels of S1P in plasma/serum, they were increased in cisplatin nephrotoxicity [208], UUO [194,202], patients with NS [83] and those with CGN [209,210]. In addition, plasma levels of HDL-S1P [210] were increased in patients with CGN and end-stage renal disease and negatively correlated with albuminuria [210]. These data suggest that HDL-S1P may protect against kidney injury associated with CGN and can be a biomarker for the severity of the disease.

A recent study reported that plasma levels of S1P were lower in SLE patients due to enhanced activation of SGPL1 as compared to controls, which inversely correlated with the disease activity and increased after immunosuppressive treatment [93]. This suggests that lower plasma levels of S1P may be a biomarker for the disease activity of SLE patients. In contrast, serum levels of S1P were higher in patients with LN and SLE compared to control, whereas no difference was noted between those with LN and SLE patients without renal injury [92]. Another study also showed that serum levels of S1P tended to be higher in patients with active LN than those without [211], although urine S1P levels remained unchanged. In addition, plasma/serum levels of both S1P and dh-S1P and the levels of dh-S1P but not S1P in the kidney were higher in LN mice [94]. Inhibition of SphK2 reduced serum levels of S1P and dh-S1P, while dh-S1P levels in the kidney of LN mice were elevated [94]. These data suggest that dh-S1P synthesis in the kidney of LN is mediated by SphK1 activity but not SphK2. Furthermore, the expression of SphK2 in peripheral blood mononuclear cells remained unchanged in lupus patients, while serum levels of S1P and dh-S1P were higher in SphK2^−/−^ lupus mice than SphK2^+/+^ lupus mice [212]. These data suggest that SphK1 contributes to a rise in serum levels of S1P and dh-S1p. In this study, depletion of SphK2 did not affect the progression of LN. Thus, both SphK1/2 contribute to a rise in serum levels of S1P in LN, while SphK1 but not SphK2 may protect against LN. However, it remains to be determined whether the levels of S1P in plasma and the kidney predict the disease activity and what a role of SphK1/2 plays in the progression of LN.

Plasma levels of S1P were increased in diabetic mice [96] and positively correlated with glucose levels and BMI in diabetic patients [213]. In contrast, plasma S1P levels were decreased in parallel with kidney dysfunction and inversely correlated with albuminuria in DN patients [214]. These data suggest that the levels of S1P in plasma and kidney may differ and depend on the presence or absence of renal injury, and that plasma levels of S1P can be a predictive biomarker for the severity of DN. In support of this, plasma levels of apolipoprotein M (apoM), a minor HDL apo and carrier for S1P, were decreased in diabetic patients [215]. In addition, apoM deficiency can increase urine S1P excretion, since apoM facilitates its renal tubular absorption [216]. Furthermore, albumin- and apoM-bound S1P can be excreted in urine, which may reduce plasma levels of S1P. These data suggest that alteration of plasma S1P levels may be dependent on its renal synthesis, tubular absorption and urine excretion in DN. Further studies are needed to examine whether these factors determine the levels of S1P in the kidney and whether the levels of plasma/serum and kidney can predict severity of the disease in oxidative stress-mediated kidney diseases.

**Table 3 ijms-23-04010-t003:** Alteration of sphingosine-1-phosphate levels and kidney cell response in oxidant-induced kidney injury.

Oxidative Kidney Diseases	S1P Levels in Kidney and Plasma/Serum		Enzymes for S1P Alteration	Kidney Cell Response	References
Toxic nephropathy Cadmium	kidney, plasma	kidney→, plasma↑	kidney SphK1/2→, S1PR1→	Necrosis/ Fibrosis	[208]
Radiation	podocyte	↓	SMPDL3b↓	Apoptosis	[35]
Ischemia/reperfusion	RTC	↓	SphK1↓	Necroptosis	[195]
	kidney	NA	SphK1↑, SphK2→, S1PR1/3↑	Apoptosis/necrosis	[196]
	EC	NA	SphK1↑, SphK2→	Angiogenesis	[197]
Unilateral ureteral obstruction	Kidney, plasma	kidney; NA, plasma↑	RTC; SphK1↑	Autophagy/fibrosis	[201]
	Kidney	→	SphK→	NA	[81]
	Kidney	kidney; NA	SphK1↑, S1PR1-3 ↑	Necrosis/fibrosis	[202]
	Kidney, plasma	↑	SphK2↑	Necrosis/fibrosis	[194]
Anti-GBM Ab GN	Spleen	NA	S1PR1/2/5 ↑	Necrosis	[203]
Nephrotic syndrome	Podocyte, serum	serum↑	SGPL1↓	FSGS	[83]
	Podocyte, fibroblast	fibroblast↓	SGPL1↓	FSGS	[84]
Chronic GN	Plasma	↑	UN	Necrosis	[209]
	Plasma	S1P↑, HDL-S1P↑	UN	ESRD	[210]
Lupus nephritis	Plasma	↓	SGPL1↑	NA	[93]
	Serum, urine	serum↑, urine→	UN	NA	[211]
	Kidney, serum	serum S1P↑, kidney, serum dhS1P↑	SphK1/2↑	Necrosis	[92,94]
	Serum	serum S1P↑, dhS1P↑	PBMC SphK2→	NA	[212]
Diabetic nephropathy	Kidney	↓	UN	Apoptosis/MME/inflammation/ fibrosis	[43]
	Kidney, glomerulus, MC	↑	SphK↑	Proliferation	[46,47]
	MC	NA	SphK1↑	Fibrosis	[198]
	Kidney	↑	SphK1↑	Necrosis/proliferation/ inflammation/fibrosis	[199]
	Kidney, RTC	↑	SphK1↑	Inflammation/fibrosis	[200]
	Kidney, MC	NA	SphK1↑	Fibrosis	[204]
	VSMC	↑	SphK1↑, SphK2→	Apoptosis	[205]
	Plasma	↑	UN	Necrosis	[96,213]
	Plasma	↓	UN	Necrosis	[214]

dh-S1P; dihydro-S1P, ESRD; end-stage renal disease, MME; mesangial matrix expansion, PBMC; peripheral blood mononuclear cell, RTC; renal tubular cell, S1PR; S1P receptor. ↑; increased, ↓; decreased, →; unchanged.

### 7.2. ROS Regulate SphKs/S1P and Vice Versa and the Role of Mitochondria for Regulation of ROS by S1P

ROS can regulate SphKs/S1P and vice versa. For example, a low H_2_O_2_ concentration activated SphK1, leading to cell proliferation, while a high H_2_O_2_ concentration was toxic by suppressing SphK1 in VSMC [217]. These data suggest that activation of SphK1 depends on the severity of oxidative stress. In addition, a high-glucose condition activated SphK1 activity in MCs [198] and VSMC [205] through its translocation into the plasma membrane, which was inhibited by antioxidants, NAC and GSH [205]. Similarly, LDL, which increased ROS production [218], promoted translocation of SphK1 from cytosol to plasma membrane, enhancing connective tissue growth factor (CTGF) expression in MCs [219]. Furthermore, serum deprivation increased ROS production [220], which suppressed SphK1 activity in HEK cells, leading to cell growth. This event was ameliorated by an antioxidant, NAC [221]. In contrast, serum deprivation-induced ROS production induced SphK2 expression and its translocation from cytosol into nuclei, leading to apoptosis and downregulation of SphK2 prevented these events in HEK cells [222]. These data suggest that ROS induce subcellular translocation of SphKs, which results in activation of the enzymes, while these events are counterbalanced by antioxidants and that SphK1 protects against ROS-induced apoptosis, whereas SphK2 has the opposite effect. Furthermore, a rheostat of ROS and antioxidants can regulate the activity of SphK1/2.

In support of this, SphK1 can suppress ROS generation. Transfection of SphK1 inhibited I/R-induced ROS production, while inhibition of SphK1 enhanced it in other types of cells [223]. Overexpression of S1PR1 or deletion of S1PR2 ameliorated high glucose-induced ROS generation in ECs [224]. Similarly, hypoxia induced ROS production in ECs, while S1P/S1PR1 suppressed hypoxia-induced ROS production [225]. These data suggest that SphK1/S1PR1 can inhibit ROS production following oxidant stimuli, whereas S1PR2 has the opposite effect. Currently, it remains unknown whether or not SphK2 can suppress ROS production in oxidant-induced kidney injury.

Mitochondria play an important role in the regulation of ROS production by S1P. S1PR1 overexpression restored mitochondrial function including MRCC, ameliorating ROS production and making RTCs resistant to cisplatin, while deletion of S1PR1 blocked these events [226]. S1P located at the inner mitochondrial membrane bound to prohibitin 2 (PHB2), which regulates mitochondrial assembly and function, preserved MRCC and MMP [227], thereby ameliorating mitochondrial ROS production. These data suggest that SphKs/S1P can inhibit ROS production by restoring mitochondrial function.

## 8. Roles of Mitochondria and Cell Signaling Pathways for S1P-Induced Cellular Function in Oxidant-Induced Kidney Injury

### 8.1. A Role of SphK1/2 for S1P-Induced Survival

Although SphK1/2 produces S1P, SphK1 generally promotes survival, while SphK2 induces apoptosis [228,229]. Transfection of SphK1 rendered HEK cells more resistant to cisplatin, while that of SphK2 had the opposite effect [109]. SphK1/S1PR1 also protected against I/R-induced apoptosis in the kidney [230] and ECs [225], while SphK1-knockout aggravated it [230]. In addition, a S1PR2 antagonist ameliorated I/R kidney injury by activating SphK1, but this effect was abolished in SphK1-deficient mice [231], while a S1PR2 agonist exacerbated it by downregulating SphK1 [231]. Furthermore, a S1PR2 antagonist activated SphK1, ameliorating H_2_O_2_- and TNF-α-induced apoptosis/necrosis in HK-2 cells, while a S1PR2 agonist had the opposite effect [231]. These data suggest that SphK1/S1PR1 protects against oxidant-induced kidney injury, whereas S1PR2 has the opposite effect by downregulating SphK1.

In contrast to the survival effects of SphK1, SphK2 promotes apoptosis/cell death in various types of oxidant-induced kidney injury. A lack of SphK2 protected against I/R-induced kidney injury [230], although it was challenged [196]. The expression of SphK2 was similar between LN patients and controls [94], and deletion of SphK2 [94,212] failed to ameliorate LN, suggesting a minor role of SphK2 in LN. However, SphK2 deficiency rendered HEK cells resistant to serum deprivation and TNF-α [222], which produce ROS/ceramide. In addition, depletion of SphK2 prevented diabetic podocyte injury [207]. Thus, an apoptotic role of SphK2 may depend on types of kidney cells and oxidant stimuli. However, these data described above strongly support the notion that SphK1 generally promotes survival, while SphK2 has the opposite effect in oxidant-induced kidney injury.

### 8.2. Role of Mitochondria for S1P-Induced Cell Survival

Mitochondria play a crucial role for S1P-induced cell survival in oxidant-induced kidney injury (Figure 2 and Figure 6) [4]. S1P/S1PR1 prevented cisplatin-induced apoptosis in RTCs [226] and hypoxia-induced apoptosis in ECs [225] by preserving MRCC, which prevents MOMP and restores mitochondrial function, thereby decreasing ROS production, the Bax/Bcl-2 ratio and Cyto C release from the mitochondria. Thus, SphK1/S1PR1 functions as a survival factor by preserving mitochondrial function which inhibits ROS production, upregulating anti-apoptotic and downregulating apoptotic Bcl-2 family proteins. The protective effect of SphK1 on mitochondrial function depends on the type of S1PRs. In support of this, overexpression of SphK1 induced upregulation of Bcl-X [232] and Bcl-2 [233], and downregulation of Bim [232] promoted survival in ECs under serum deprivation, which produces ROS and ceramide.

In contrast, the expression of SphK2 enhanced cisplatin nephropathy [109]. SphK2 localized at the mitochondria may result in the activation of Bak and release of Cyto C from the mitochondria [139]. Overexpression of SphK2 promoted apoptosis in β cells by interacting with Bcl-xL, contributing to the progression of DN [234]. Transfection of SphK2 also rendered MCs and RTCs more sensitive to TNF-α-induced mitochondrial dysfunction which produces ROS and ceramide, by decreasing Bcl-xL [53]. In addition, the cooperation of SphK2 with C8-BID stimulated the expression of Bax and MOMP, leading to apoptosis. This event was inhibited by Bcl-xL [139]. Furthermore, SphK2 contained BH3-only proteins, a pro-apoptotic subgroup of the Bcl-2 family, and interacted with Bcl-xL, leading to suppression of its anti-apoptotic effects in oxidant-induced apoptosis of HEK cells [228]. These data suggest that SphK2 promotes apoptosis by upregulating apoptotic Bcl-2 family proteins and by downregulating anti-apoptotic Bcl-2 family proteins in oxidant-induced kidney injury.

In addition, upon oxidative stimuli, SphK2 moved from the cytosol to the ER and induced Ca^2+^ release from the ER, which was dispensable for SphK2 activation in HEK cells [229]. In support of this, mitochondrial Ca^2+^ uptake from the ER due to ER stress induced by H_2_O_2_ and ceramide was dispensable for SphK2-induced apoptosis, leading to Bax/Bak and caspase activation [229]. Thus, an increase in mitochondrial Ca^2+^ uptake from ER by SphK2, which is dependent on Bax/Bak, is crucial for SphK2-induced apoptosis. Taken together, these lines of evidence suggest that SphK1 promotes survival, while SphK2 has the opposite effect in oxidant-induced kidney injury.

Furthermore, hexadecenal, a degraded product of S1P by S1PL, stimulated ROS production, Bax, Bid cleavage and mitochondrial Bim translocation, leading to apoptosis in HEK cells, while these were blocked by NAC [235]. Thus, a ROS-antioxidant rheostat is crucial in maintaining basal S1P levels for cellular decision to trigger or inhibit apoptosis.

### 8.3. Cell Signaling Pathways for S1P-Induced Cell Survival, Proliferation, Inflammation and Fibrosis

#### 8.3.1. Apoptosis

During hypoxia, accumulated ROS stimulated the expression of hypoxic homeostasis transcription factors such as hypoxia-inducible factor-1α (HIF-1α) and heat-shock protein 27 (HSP27) (Figure 7) [236]. Hypoxia activated SphK1 in ECs through the activation of SphK1 promoter activity which contains two putative HIF-responsive-elements (HREs), contributing to SphK1 gene transcription [205]. Thus, deletion of HREs abrogated hypoxia-induced SphK1 promoter activity and deficiency of SphK1 abolished hypoxia-induced angiogenesis [205]. In addition, a selective A(1) adenosine receptor (AR) agonist [237] and IL-11 [238] protected I/R-induced kidney injury and apoptosis/necrosis in RTCs by activating SphK1 via nuclear translocation of HIF-1α, whereas the inhibition of HIF-1α blocked SphK1 activity [238]. On the contrary, the protective effect of AR agonist and IL-11 on I/R-induced kidney injury was abolished in SphK1-deficient mice [237,238]. Furthermore, overexpression of SphK1 prevented I/R-induced apoptosis/necrosis in the kidney and H_2_O_2_-induced apoptosis in HK-2 cells by increasing HSP27 expression, while this effect was blocked by S1PR1 antagonism [230,239]. Taken together, these data suggest that HIF-α activates SphK1 activity and SphK1/S1P1R-induced HSP27 protects against apoptosis in oxidant-induced kidney injury. ERK protects against apoptosis and can activate another survival factor, SphK1, in HEK cells [240]. S1PR1 agonist ameliorated oxidant-induced kidney injury such as lipopolysaccharide (LPS)- and I/R-induced kidney injury and apoptosis by activating ERK/Akt [241]. Hydrogen peroxide-induced EC apoptosis was ameliorated by activating ERK and SphK1, while inhibition of SphK1/S1PR1 abolished this effect [242], suggesting that SphK1 inhibits ROS-induced apoptosis by activating ERK. Thus, SphK1 and ERK activate each other and function as anti-apoptotic in oxidant-induced kidney injury.

In contrast, MCs overexpressing SphK2 were susceptible to apoptosis caused by LPS and TNF-α, which produce ROS and ceramide by inhibiting ERK/Akt/protein kinase B (PKB) and Bcl-xL [53]. In addition, S1P/S1PR1 activated Akt and eNOS which counteracted ROS production, ameliorating hypoxia-induced EC apoptosis [225]. Inhibition of either ERK or PI3K alone did not affect cisplatin sensitivity in cells overexpressing S1PL which degrade S1P and thus decreases S1P levels, whereas simultaneous inhibition of ERK and PI3k increased cisplatin sensitivity [243]. Thus, ERK/PI3K/Akt/Bcl-xL function as survival factors in ceramide-induced apoptosis. In addition, SphK2 promotes apoptosis by inhibiting ERK/PI3K pathway, while SphK1 has the opposite effect.

IGF-1 inhibited hyperglycemia-induced ROS production [244] and prevented hyperglycemia-induced apoptosis by activating Akt/PKB in MCs [245]. SphK1 activated IGF/IGF binding protein (IGFBP)-3, which in turn stimulated downstream ERK/Akt activity, thereby ameliorating ROS-induced apoptosis in ECs [246]. Conversely, IGF-1 activated SphK1 and this effect was mediated by ERK/PI3K/Akt in ECs [247]. Furthermore, H_2_O_2_ downregulated VEGF via inhibition of theh PI3K/Akt pathway, leading to apoptosis in RTCs [248]. However, SphK1-mediated IGFBP-3 activated VEGF, which resulted in angiogenesis [249]. Thus, IGF and VEGF promote survival by activating survival ERK/PI3K/Akt signaling pathways. Taken together, these data suggest that SphK1 protects against apoptosis by activating ERK/PI3K/Akt/PKB and growth factors such as IGF and VEGF which share the same survival signaling pathways in oxidant-induced kidney injury.

On the other hand, overexpression of SphK2 in CerS1-expressing HEK cells [109] and of S1P lyase resulted in decreased S1P levels in HEK cells [243] and activated p38MAPK, leading to increased sensitivity to cisplatin. Transfection of SphK1 did not abrogate cisplatin-induced p38MAPK activation but activated the survival ERK, which is antagonistic to p38MAPK, thereby increasing resistance to cisplatin [109]. Thus, SphK1 does not affect p38MAPK, but activates ERK/PI3K which counteracts p38MAPK, while SphK2 activates p38MAPK, leading to apoptosis. In fact, S1P ameliorated H_2_O_2_-induced EC apoptosis by inhibiting p38MAPK but not JNK [249]. OxLDL activated ROS production and p38MAPK, resulting in ceramide formation and apoptosis in ECs, and inhibition of SphK1 enhanced these events [34]. These data suggest that SphK1 inhibits p38MAPK activity, leading to survival, while SphK2 activates it, leading to apoptosis, and thus p38MAPK functions as apoptotic in oxidant-induced kidney injury.

As another MAPK, hexadecenal, a degraded product of S1P, activated JNK but not ERK/Akt/p38MAPK, leading to apoptosis in HEK cells, while these were blocked by NAC [235]. This indicates that antioxidants maintain S1P levels which inhibit JNK activity and ameliorate apoptosis. Finally, ischemia induced apoptosis/necrosis in NRK-52E cells by activating PP2A [188], which inhibits SphK1 [250]. These data suggest that SphK1/S1P inhibits JNK and PP2A, thereby inhibiting apoptosis in oxidant-induced kidney injury.

#### 8.3.2. Cell Proliferation

S1P-induced signaling pathways regulate cell proliferation, inflammation and fibrosis in oxidant-induced kidney injury (Figure 8). First of all, SphK/S1P plays an important role in the regulation of cell proliferation in the kidney through a variety of downstream signaling pathways following oxidant stimuli. SphK1 promotes cell proliferation, while SphK2 functions as anti-proliferative [50]. AGEs or H_2_O_2_ stimulated the release of dipeptidyl peptidase-4 (DPP-4) from ECs, a type II transmembrane glycoprotein expressed at various cells including MCs, which produces superoxide [251]. This event was blocked by mannose 6-phosphate (M6P)/insulin-like growth factor II receptor (M6P/IGF-IIR) signaling and an antioxidant, NAC [251]. Binding of IGF-II to M6P/IGF-IIR activated ERK through PKC-mediated SphK1 activity, leading to MC proliferation [252]. Similarly, hyperglycemia/AGEs activated SphK1/S1P, leading to glomerular [43] and MC proliferation via TGF-β [253]. These data suggest that IGF-induced SphK1 activates ERK in a PKC-dependent manner, while SphK1 by itself activates the TGF-β signaling pathway, and that these events promote cell proliferation in oxidant-induced kidney injury.

Cells decide to be proliferative or anti-proliferative under pathological conditions following sensing the severity of oxidant stimuli. Low concentrations of H_2_O_2_ activated nSMase2 through src/platelet-derived growth factor receptor (PDGFR)β/SphK1, in which PDGFRβ was required for SphK1 activation [217]. This process promoted VSMC proliferation. In contrast, high concentrations of H_2_O_2_ inhibited SphK1 activity and its toxicity was similar between control and nSMase2-deficient VSMC [251]. Thus, src/PDGFRβ/SphK1 pathway functions as proliferative at low oxidative stress, and the inhibition of this pathway may be anti-proliferative and toxic at severe oxidative stress. SphK1/S1P also promoted PDGF-induced MC proliferation by activating ERK, while the inhibition of SphK1 reduced it [254]. In addition, PDGF activated SphK1 activity and SphK1/S1P cooperates with PDGF to promote MC proliferation [254]. Thus, SphK1 activates PDGF by activating ERK, leading to cell proliferation. Furthermore, overexpression of COX-2 reduced ROS production, leading to the inhibition of PDGF-induced MC proliferation, whereas this effect was abolished by an increase in intracellular ROS levels [255]. These data suggest a negative link between PDGF and COX-2 in the regulation of SphK1-induced MC proliferation.

In contrast, the overexpression of SphK2 reduced MC proliferation by inhibiting ERK/Akt/PKB [53]. SphK2 knockout increased the expression of SphK1/S1PR3 and ERK/PI3K/Akt due to an increase in SphK1 activity, leading to increased proliferation/migration, whereas inhibitors of ERK and of S1PR3 abolished SphK1-induced MC proliferation [256].

Taken together, these data suggest that SphK1 promotes cell proliferation via ERK/PI3K/Akt, PDGF and TGF-β, whereas SphK2 has the opposite effect by inhibiting ERK/PI3K/Akt. Further studies are needed to determine whether SphK1 and SphK2 differentially regulate cell proliferation in oxidant-induced kidney injury.

#### 8.3.3. Inflammation

Cisplatin induced proinflammatory cytokines (TNF-α, IL-6, etc.) and infiltration of neutrophils/macrophages in the kidney, while these were attenuated by a S1PR1 agonist, FTY720, in wild type mice but not in S1PR1-knockout mice [226]. A S1PR1 agonist, SEW2871, also ameliorated I/R kidney injury by reducing the infiltration of neutrophils/macrophages and proinflammatory molecules (TNF-α, P-selectin, E-selectin and intercellular adhesion molecule-1; ICAM-1) [257]. These data suggest a protective role of S1P/S1PR1 for inflammation in oxidant-induced kidney injury.

Recent evidence suggests that SphK1 protects against renal inflammation, while SphK2 has the opposite effect in oxidant-induced kidney injury. For example, downregulation of the S1P transporter, spinster homologue 2 (Spns2) located in RTCs, which exports S1P and reduces S1P levels, inhibited TNF-α- and IL-1β-induced monocyte chemotactic protein (MCP)-1 expression in HK-2 cells, while SphK1-knockdown blocked these events [258]. Similarly, in SphK1-knockout HK-2 cells, the expression of MCP-1 was enhanced in response to inflammatory cytokines as compared to the control. These data suggest that SphK1 protects against renal inflammation by inhibiting downstream pathways of TNF-α- and IL-1β. In support of this, expression of proinflammatory markers (Cxcl2, Tlr4, MCP-1, Pcna, and Col3a1) was increased in the kidneys of wild type and Sphk1^−/^^−^ mice but not in Sphk2^−/^^−^ mice, suggesting a protective effect of SphK1 on renal inflammation [259]. Furthermore, SphK1 prevented I/R-induced inflammation via the activation of HIF-1α [238] and HSP27 [230]. Taken together, these data suggest that SphK1 prevents inflammation via HIF-1α and HSP27 and by inhibiting proinflammatory cytokines in oxidant-induced kidney injury.

In contrast, deletion of SphK2 decreased neutrophil infiltration in I/R kidney injury [259], inflammatory macrophage infiltration [204], TGF-β1 expression, inflammatory cytokines (MCP-1,TNF-α, chemokine ligands-1; CXCL1, and IL-1β) and increased anti-inflammatory (M2) macrophage infiltration in UUO kidney [260], ameliorating renal inflammation. In addition, SphK2 with Fyn promoted inflammation via STAT3/Akt, while SphK2 knockout reduced it by inhibiting Fyn-STAT3/Akt but not TGF-β1 [54]. Thus, SphK2 promotes inflammation by activating TGF-β1, proinflammatory cytokines and Fyn-STAT3/Akt pathway. Taken together, these data suggest that SphK1 protects against renal inflammation, while SphK2 promotes it in oxidant-induced kidney injury.

#### 8.3.4. Renal Fibrosis

In I/R kidney injury, renal fibrosis was similar between wild and SphK1-knockout mice, suggesting a minor role of SphK1 for I/R-induced fibrosis [259]. However, S1P promoted fibrosis in UUO [261]. The expression of SphK1/S1PR1-3 was increased in UUO kidney [196,202,262] and SphK1-knockout protected RTCs [262] and the kidney [202] from fibrosis in UUO by inhibiting NF-κB. As another factor, TGF-β produced ROS and vice versa and induced fibrosis [263] via the activation of SphK1/S1P [262], although it was challenged [201].

Berberine, an alkaloid from a Chinese herb that has an anti-oxidative effect, suppressed high glucose-induced α-SMA, FN, TGF-β1 and AP-1 by inhibiting SphK1 activity in MCs [264]. SphK1/S1P upregulated FN and TGF-β1 through the activation of S1PR2 and NF-κB in diabetic MCs [198,264] and promoted DN-induced fibrosis [204,253], while the inhibition of SphK1/S1PR2 prevented these events [264]. In addition, hyperglycemia stimulated TGF-β-induced FN expression in HK-2 cells, while the inhibition of SphK1 prevented these events by inhibiting ERK/AP-1/NF-κB [198]. Furthermore, SphK1 phosphorylated casein kinase 2α (CK2α), while CK2α-knockdown suppressed SphK1-induced NF-κB activation [204]. However, SphK1-knockdown diabetic mice suppressed fibrosis via the inhibition of CK2α/NF-κB [204]. Taken together, these data suggest that SphK1 promotes renal fibrosis via TGF-β/ERK/AP-1/NF-κB/CK2α pathways and by inhibiting PP2A in oxidant-induced kidney injury.

On the other hand, SphK2-knockout ameliorated I/R- [259] and UUO-induced renal fibrosis [88,260] by inhibiting TGF-β1. By contrast, the overexpression of SphK2 exacerbated fibrosis in UUO kidney by decreasing Smad7 [194]. These data suggest that SphK2 promotes fibrosis through the TGF-β-dependent pathway. Furthermore, SphK2, together with Fyn, promoted fibrosis via signal transducer and activator of transcription 3 (STAT3)/Akt and SphK2-knockout mice exhibited lower levels of extracellular matrix in the kidney by targeting SphK2-Fyn-STAT3/Akt but not TGF-β1 [54]. Thus, SphK2 promotes fibrosis by means of TGF-β and the Fyn/STAT3/Akt pathway. Taken together, these data suggest that both SphK1 and SphK2 promote renal fibrosis through different downstream signaling pathways in oxidant-induced kidney injury.

## 9. A Rheostat of Ceramide-S1P in the Regulation of Oxidative Stress-Mediated Kidney Injury

### 9.1. Balance between Ceramide and S1P Regulates Oxidant-Induced Kidney Injury

As discussed earlier, ceramide and S1P have the opposite effect on cellular functions in general. This implies that interaction between the enzymes that generate ceramide and S1P, which modulates ceramide-S1P rheostat, can determine the cell fate following oxidant stimuli (Figure 2). The following observations support this hypothesis. A S1P agonist, FTY720, inhibited CerS activity in HEK cells [265]. Overexpression of S1P phosphatase degraded S1P, resulting in decreased S1P levels and maintained ceramide levels, which resulted in an increase in the ratio of ceramide/S1P, leading to apoptosis in HEK cells exposed to oxidative stress/ceramide [266]. Studies with gene engineering showed that knock-down of SphK1/S1P increased ceramide production by de novo ceramide synthesis or the salvaging pathway in RTCs [267,268], which resulted in an increased ratio of ceramide/S1P, leading to apoptosis and the inhibition of cell proliferation. In contrast, the overexpression of SphK1 increased S1P but decreased ceramide levels in HEK cells [269]. In addition, HEK cells transfected with Ras oncogene, K-RasG12V, which activates Raf/MEK/ERK, increased SphK1/S1P levels but decreased ceramide production in a SphK1-dependent manner [270], suggesting that Raf/MEK/ERK may regulate a ceramide-S1P rheostat. On the other hand, overexpression of SphK2 in HEK cells increased the incorporation of palmitate, a substrate for both PST and CerS, into C16-ceramide, whereas SphK1 decreased it [270], leading to reduced ceramide-S1P rheostat and apoptosis. These data suggest that there is an interaction between ceramide-producing enzymes and SphKs and that SphK1 may increase while SphK2 decreases the ratio of ceramide/S1P, leading to differential regulation of oxidant-induced kidney injury. In support of this, SphKs can regulate the activity and nature of ceramide-producing enzymes, which in turn change their regulation of anti-apoptotic signaling pathways from the original apoptotic downstream signaling pathways. The inhibition of SphK (although which SphK, SphK1 or SPhK2, is unknown) but not Degs1 induced the polyubiquitination of Degs1 via a mechanism involving oxidative stress, p38MAPK and Mdm2 (E3 ligase) in HEK cells [17]. This form of Degs1 changes its function from a pro-apoptotic form to pro-survival. This observation indicates that a rheostat of activity of the ceramide-producing enzymes and SphKs may modulate their downstream signaling pathways for apoptosis or survival in oxidant-induced kidney injury.

A rheostat of ceramide-S1P plays a crucial role in the regulation of apoptosis in various types of oxidant stress-mediated kidney diseases. Transfection of SphK1 into RTCs overexpressing CerS1 rendered the cells more resistant to cisplatin toxicity [108], suggesting that a rheostat of ceramide-S1P can determine sensitivity to cisplatin. Tunicamycins, antibiotics which induce ROS production and ER stress [271], increased the ceramide/S1P ratio, leading to necrosis in the kidney [272]. In addition, radiation-induced ceramide production caused apoptosis in ECs, whereas the treatment with S1P prevented these events by decreasing the ratio of ceramide/S1P [68]. SMPDL3b-overexpressing podocytes had higher basal S1P levels and maintained basal ceramide levels, which resulted in decreased ceramide-S1P rheostat, and were protected from radiation-induced cytoskeletal remodeling [35]. Furthermore, administration of bone marrow-derived mononuclear cells (BMMC) decreased ceramide and increased S1P levels, ameliorating UUO-induced kidney injury [81].

In a model of hypertensive CKD with angiotensin II (Ang II) infusion, where ROS is involved, renal hypoxia, hypertension, proteinuria and fibrosis were more severe in Ang II-infused erythrocyte-specific SphK1 knockout (*eSphK1^−/−^*) mice compared with controls [273]. Increased erythrocyte S1P activates AMP-activated protein kinase (AMPK) 1α and bisphosphoglycerate mutase (BPGM) by reducing the ceramide/S1P ratio, leading to increased Hb production and thus more O_2_ delivery to counteract kidney hypoxia and progression to CKD [270]. This implies that more S1P levels in erythrocytes reduce the ceramide/S1P ratio, thereby ameliorating kidney injury associated with CKD by inhibiting kidney tissue hypoxia. In addition, the fibroblasts from NS patients with mutation of SGPL1 showed an increase in very long-chain ceramides due to CerS2 activation, whereas S1P levels were below the limits of detection [75], suggesting that a rheostat of ceramide-S1P may contribute to the maintenance of podocyte function in NS. An increased ratio of serum levels of ceramide/S1P could discriminate SLE patient groups with higher activity of the disease [92,93]. Furthermore, the ratio of 24.1ceramide/S1P in serum and plasma was significantly higher in SLE patients with renal injury than in those without, indicating that ceramide-S1P rheostat in serum/plasma can be a biomarker of renal impairment in SLE patients [92]. Taken together, these data indicate that a high ceramide/S1P ratio determines the severity of oxidant-induced kidney injury and can be a biomarker of renal impairment.

A rheostat of ceramide-S1P can regulate islet cell viability and function [274] as well as insulin sensitivity, contributing to the development of DM/DN. Ceramide induced ROS production, leading to islet β-cell dysfunction which resulted in DM [275], while S1P protected it [276]. Overexpression of SphK1 in high-fat diet-fed mice increased SphK1 activity in skeletal muscle, accompanied by decreased intramuscular ceramide accumulation, thereby inducing insulin sensitivity [277]. In addition, AdipoRon lowered cellular ceramide levels by activation of aCDase, which normalized the ceramide/S1P ratio, ameliorating albuminuria and lipid peroxidation in DN [43]. Furthermore, low concentrations of AGEs activated nCDase and SphK activity in MCs, resulting in a decreased ceramide/S1P ratio, which led to MC proliferation, whereas higher concentrations of AGEs had the opposite effect [47]. Moreover, insulin treatment corrected a ceramide/S1P ratio in diabetic mice [96]. These lines of evidence suggest that ceramide-S1P rheostat plays a crucial role in the pathogenesis of DM/DN, and thus modulators of SphK/S1P are currently under development for treatment of the disease [278]. 

Finally, a rheostat of ceramide-S1P can regulate endothelial and vascular function which promotes progression of oxidant-induced kidney injury. Balance between oxLDL-induced survival and apoptotic responses was dependent on a ceramide-S1P rheostat, which regulated endothelial function, angiogenesis and atherosclerosis [279]. Macrophages from SphK2-knockout mice showed an increase in ceramide levels, while SphK1 overexpression in SphK2-deficient mice reduced ceramide levels, thereby ameliorating atherosclerosis [280]. Similarly, ceramide-S1P rheostat regulated the transition from NO-dependent to H_2_O_2_-mediated flow-induced dilation of arterioles, a microvascular dysfunction, which is predictive of cardiovascular events and atherosclerosis [281]. In addition, sensing the severity of oxidant stress in the cells can determine a rheostat of ceramide-S1P. Low H_2_O_2_ concentrations activated nSMase2 and ShK1 through a nSMase2/ceramide-dependent signaling pathway that acts upstream of SphK1, leading to cell proliferation, while high H_2_O_2_ concentrations inhibited SphK1 activity and reduced the ratio of ceramide/S1P, leading to cytotoxicity in VSMC [217]. In support of this, a rheostat of ceramide-S1P has been shown to control the blood pressure in hypertensive patients [282]. Thus, sensing the severity of oxidative stress in the cell can determine a rheostat of ceramide-S1P, which plays a crucial role in the regulation of oxidant-induced kidney injury.

### 9.2. Targeting Enzymes for Ceramide Generation and SphKs That Improve Ceramide-S1P Rheostat, Contributing to Prevention against Oxidant-Induced Kidney Injury

Targeting enzymes that generate ceramide and S1P to restore a rheostat of ceramide-S1P have been considered as a therapeutic strategy in various diseases (reviewed in [283,284]). In the same line, the application of compounds that can reduce the ratio of ceramide/S1P by targeting the enzymes involved in ceramide production and SphKs would be beneficial to prevent oxidant-induced kidney injury (Table 4) [27,43,62,66,71,74,75,76,78,85,94,198,200,207,219,226,260,264,281,284,285,286,287,288,289,290,291,292,293]. Among these compounds, fumonisin B1, for the treatment of acute rejection, where ROS are involved, and fingolimod (FTY720) have only been used in kidney transplant recipients [294,295].

As a tool for targeting ceramide-producing enzymes and SphKs, experimental data show that antioxidants would be beneficial as the treatment of oxidant-induced kidney injury by restoring ceramide-S1P rheostat. As discussed earlier, antioxidants inhibit ceramide-producing enzymes, and conversely the inhibition of antioxidants by ROS activates these enzymes (Section 4.3 and Section 4.4). In addition, overexpression of antioxidants [142,143,144] reduced apoptotic-Bcl-2 family proteins that activate ceramide-producing enzymes and increased anti-apoptotic Bcl-2 family proteins that inhibit ceramide-producing enzymes (Section 5.2). Furthermore, a rheostat of ROS-antioxidants can regulate the activity of SphKs (Section 7.2). In support of this, adiponectin [284] and coumestrol [296] that function as antioxidants and administration of antioxidants [297] reduced the ratio of ceramide/S1P, contributing to prevention against oxidative stress-mediated diseases (Table 4). As described earlier, the binding of adiponectin, an antioxidant, to its receptor, AdipoR, further enhances the receptor’s CDase activity [12]. In addition, AdipoRon, an adiponectin receptor agonist, reduced ceramide levels by activating aCDase activity and normalized the ceramide/S1P ratio in podocyte [298] and GECs [43], ameliorating apoptosis of these cells in DN and ROS-induced endothelial dysfunction in arterioles from patients with coronary artery disease [281]. Thus, the application of antioxidants can improve ceramide-S1P rheostat under oxidative stress, thereby ameliorating oxidant-induced kidney injury.

Finally, glucocorticoid, used as a treatment for CKD, has been beneficial as a therapy for LN by decreasing plasma ceramide/S1P ratio [93], although the target enzymes remain unknown. In addition, glucocorticoids protected against oxidant stress-induced MC apoptosis by upregulating nCDase and SphK, which reduced ceramide/S1P ratio [299] (Table 4).

Taken together, these data suggest that targeting the enzymes that regulate ceramide formation and SphKs and application of antioxidants to improve a rheostat of ceramide-S1P would be new therapeutic strategies to prevent oxidative-stress-mediated kidney diseases.

## 10. Conclusions

ROS play a pathogenic role in kidney injury by regulating SL metabolism. ROS increase ceramide levels but inhibit S1P formation, while antioxidants have the opposite effect. Conversely, ceramide induces ROS production and inhibits antioxidants, while S1P has the opposite effect. A rheostat of ROS-antioxidants and the interaction between the enzymes that generate ceramide and S1P determine the role of the ceramide-S1P rheostat in oxidant-induced kidney injury. Ceramide induces mitochondrial dysfunction including the upregulation of apoptotic Bcl-2 family proteins and downregulation of anti-apoptotic Bcl-2 family proteins, which results in MPTP opening, MOMP, the formation of ceramide channels and closure of VDAC in the mitochondria, ROS generation, perturbation of Ca^2+^ homeostasis and the ER stress. These events subsequently induce mitochondrial Cyto C release and caspase activation, leading to apoptosis. In contrast, the SphK1/S1P axis has the opposite effect to ceramide in terms of the regulation of mitochondrial function and downstream cell signaling pathways which promote survival. Although both SphK1 and SphK2 can produce S1P, SphK1 generally promotes survival, while SphK2 has the opposite effect. SphK1 and SphK2 differentially regulate cell proliferation and inflammation, whereas both promote renal fibrosis. Since ceramide and S1P differentially regulate mitochondrial function and cell signaling pathways, ceramide-S1P rheostat plays an important role in the regulation of oxidant-induced kidney injury. Further research is needed to clarify the mechanism by which ROS and antioxidants regulate the enzymes involved in SL metabolism, including the role of structure-based molecules of the enzymes and their downstream signaling pathways. In addition, agents that target the structure and nature of the enzymes to restore abnormalities of SLs metabolism including ceramide/S1P ratio and could be clinically used without systemic side effects should be facilitated in the near future.

## Figures and Tables

**Figure 1 ijms-23-04010-f001:**
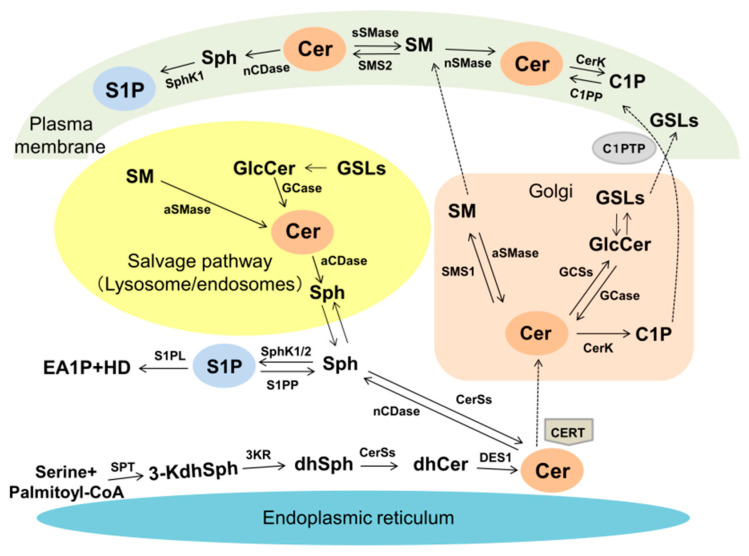
Simplified metabolic pathways of sphingolipids. Cer is de novo synthesized at the surface of the ER by condensation of serine and palmitoyl-CoA mediated by SPT, forming 3-KdhSph. 3-KdhSph is then reduced to dhSph by 3-KR. DhSph is the substrate of CerS, forming dhCer which is converted into Cer by DES1. Cer is transferred to the Golgi by CERT or vesicular trafficking and converted into SM by SMS or C1P by CerK. Cer is converted into GlcCer by GCSs and further metabolized into complex GSLs. Cer is also phosphorylated into C1P by C1PP. GSLs and SM in the Golgi are transferred to the plasma membrane by vesicular trafficking, where SM is converted into Cer by sSMase or nSMase. C1P is transferred to the plasma membrane by C1PTP. In the plasma membrane, Cer can be produced from SM via sSMase and converted into Sph by nCDase or C1P via C1PP. Sph can be metabolized into S1P by SphKs. SM in the plasma membrane enters into the recycling pathway in the acid compartment of the endolysosome, where aSMase and GCase produce Cer, which is hydrolyzed into Sph by aCDase. Once released into the cytosol, Sph is reused for Cer synthesis by CerS or phosphorylated by SphKs to yield S1P. S1P is hydrolyzed back into Sph via S1PP or degraded by S1PL into EA1P and HD. Abbreviations; aCDase; acid ceramidase, aSMase; acid sphingomyelinase, Cer; ceramide, CerK; Cer kinase, CerS; Cer synthase, CERT; ceramide transport protein, C1P; Cer-1-phosphate, C1PP; C1P phosphatase, C1PTP; C1P transfer protein, DES; dihydroceramide desaturase, dhCer; dihydroceramide, dhSph; dihydrosphingosine, EA1P; ethanolamine-1-phosphate, ER; endoplasmic reticulum, GCase; glycosidase, GCSs; glycosylceramide synthases, GlcCer; glucosylceramide, GSLs; glycosphingolipids, HD; hexadecenal, 3-KdhSph; 3-ketodihydrosphingosine, 3-KR; 3-KdhSph reductase, nSMase; neutral sphingomyelinase, SM; sphingomyelin, SMS; SM synthase, sSMase; secretaory SMase, Sph; sphingosine, SphK; sphingosine kinase, S1P; sphingosine-1-phosphate, S1PL; S1P lyase, S1PP; S1P phosphatase, SPT; serine palmitoyltransferase. Dashed arrows indicate transportation to the plasma membrane.

**Figure 2 ijms-23-04010-f002:**
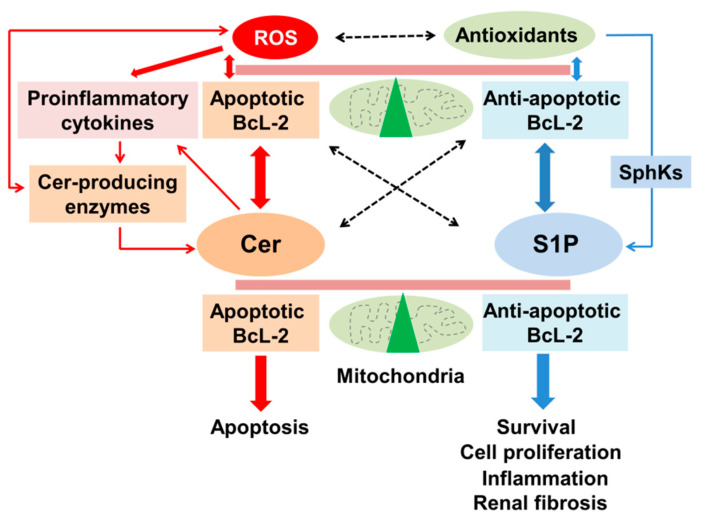
Balance between ROS and antioxidants determines ceramide-S1P rheostat that regulates oxidant-induced kidney injury. A rheostat of ROS and antioxidants regulates enzymes that generate Cer and S1P directly or through the modulation of subcellular translocation of the enzymes and upstream signaling pathways, mitochondrial function including apoptotic/anti-apoptotic Bcl-2 family proteins, and proinflammatory cytokines. ROS activate Cer-producing enzymes and vice versa, while antioxidants inhibit these enzymes. ROS stimulate proinflammatory cytokines, which in turn induce Cer generation, mitochondrial dysfunction, upregulation of apoptotic and downregulation of anti-apoptotic Bcl-2 proteins, leading to apoptosis, while antioxidants have the opposite effect. Cer induces ROS production, mitochondrial dysfunction and increased ratio of apoptotic/anti-apoptotic Bcl-2 proteins and inhibition of antioxidants, leading to apoptosis. ROS inhibit SphK/S1P and vice versa, while antioxidants activate it. SphK/S1P has the opposite biological effects to Cer, including restoration of mitochondrial function and the ratio of apoptotic/anti-apoptotic Bcl-2 family proteins, thereby promoting survival, cell proliferation, inflammation and renal fibrosis. →: stimulation, dashed arrow: inhibition. Red arrow; apoptotic, blue arrow; anti-apoptotic.

**Figure 3 ijms-23-04010-f003:**
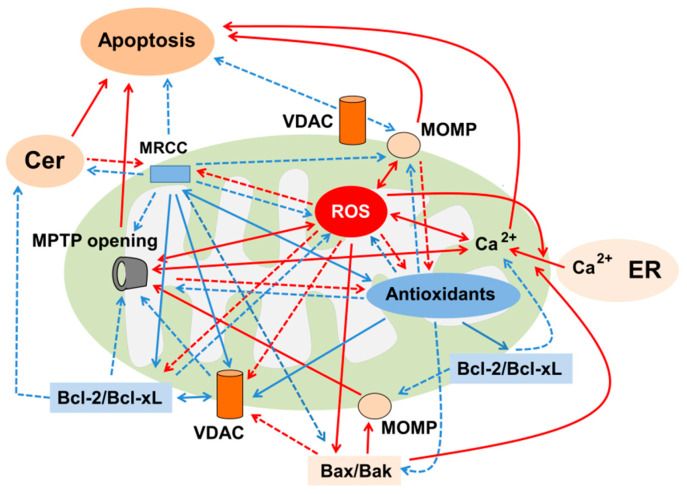
Interaction between ROS/antioxidants and mitochondria in the regulation of apoptosis. Oxidant stress inhibits MRCC, which induces ROS production. ROS and antioxidants inhibit each other. ROS inhibit MRCC, resulting in Cer production, MPTP opening, MOMP, decreased VDAC expression, suggestive of VDAC closure, and a decreased Bcl-2/Bax ratio, leading to apoptosis, while antioxidants and Bcl-2 prevent these events. Cer inhibits MRCC, leading to ROS production. Both ROS and Cer activate Bax/Bak, which enhances the mitochondrial uptake of Ca^2+^ by enhancing the transfer of Ca^2+^ from the ER, and these events trigger MPTP opening, leading to apoptosis. ROS increase intracellular Ca^2+^, which further enhances ROS production, while Bcl-2 inhibits this event and apoptosis by increasing the capacity of mitochondria to store and buffer Ca^2+^. Inhibition of MPTP opening and MOMP prevents ROS production and restores antioxidant levels. In contrast, antioxidants restore MRCC and loss of MMP, increase the expression of VDAC and Bcl-2/Bax ratio, and prevent an increase in intracellular Ca^2+^ which induces loss of MMP, ameliorating apoptosis. Antioxidants prevent an ROS-induced increase in intracellular Ca^2+^ by increasing Bcl-2 expression, which prevents a rise in intracellular Ca^2+^ by increasing the capacity of mitochondria to buffer Ca^2+^, ameliorating apoptosis. Anti-apoptotic Bcl-2 functions as an antioxidant and inhibits ROS-induced Cer formation. Abbreviations; Cer; ceramide, MMP; mitochondrial membrane potential, MOMP; mitochondrial outer membrane permeability, MPTP; mitochondrial transition pore, MRCC; mitochondrial respiratory chain complex, ROS; reactive oxygen species, VDAC; voltage-dependent anion channel. Red arrow; apoptotic pathway, blue arrow; anti-apoptotic pathway, →; stimulation, dashed arrow; inhibition.

**Figure 4 ijms-23-04010-f004:**
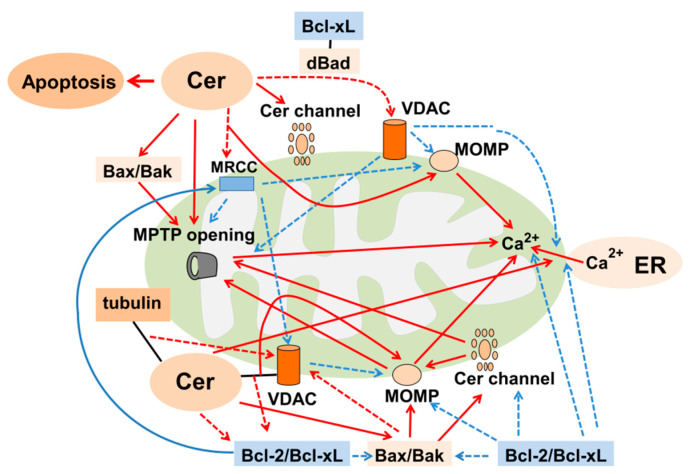
Mitochondria regulate ceramide-induced apoptosis. Cer inhibits MRCC and reduces the Bcl-2/Bax ratio, resulting in MPTP opening and MOMP, which are inhibited by Bcl-2. Cer together with Bax induces MPTP opening and MOMP, which sensitize mitochondria to Ca^2+^, leading to apoptosis. Cer induces release of Ca^2+^ from the ER, resulting in increased mitochondrial uptake of Ca^2+^, leading to apoptosis, while Bcl-2 prevents it. Bcl-2 also increases mitochondrial capacity to store and buffer Ca^2+^, thereby preventing apoptosis. Cer-induced increase in the ratio of Bax/Bcl-2 results in mitochondrial dysfunction and mitochondrial Ca^2+^ uptake, leading to apoptosis. Cer forms Cer channels. Bax/Bak enhances Cer channel formation, triggering MPTP opening and MOMP, leading to apoptosis, while Bcl-2/Bcl-xL disassembles Cer channels. Binding of Cer to VDAC induces MOMP and decreases Bcl-2/Bcl-xL expression. Cer dephosphorylates Bad (dBad) which is required for Cer-induced sensitization of MPTP opening. More Bad and less VDAC are associated with Bcl-xL at the MOM, which sensitizes MPTP to Ca^2+^, leading to apoptosis. Cer binds to tubulin and this formation induces VDAC closure, leading to mitochondrial dysfunction. Abbreviations; dBad; dephosphorylated Bad.—binding, red arrow; apoptotic pathway, blue arrow; anti-apoptotic pathway. →; stimulation, dashed arrow; inhibition. Black bar indicates binding of dBad to Bcl-xL.

**Figure 5 ijms-23-04010-f005:**
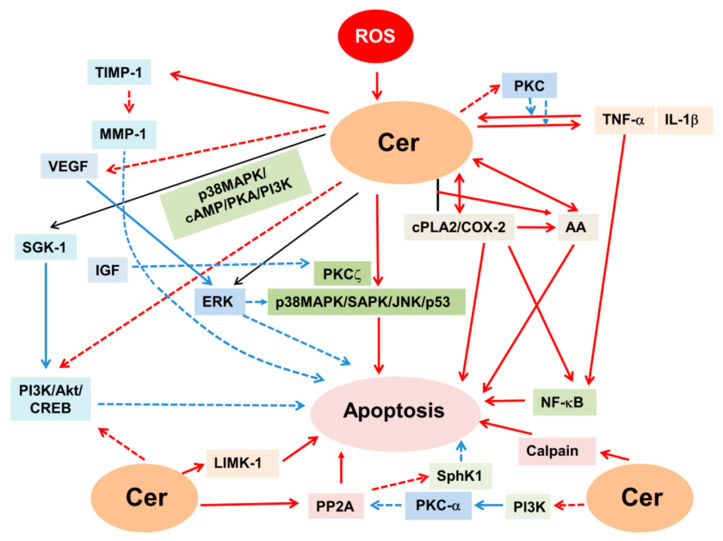
Ceramide-induced apoptotic signaling pathways in oxidant-induced kidney injury. ROS-induced Cer activates proinflammatory cytokines (TNF-α, IL-1β), ERK, p38MAPK, SAPK/JNK through PKCζ, p53, Ca^2+^-dependent calpain, NF-kB through cPLA2/COX-2, and TIMP-1, which decreases MMP-1 that suppresses apoptosis, leading to apoptosis. All these factors function as apoptotic except for anti-apoptotic ERK, which is antagonistic to p38MAPK activation, and MMP-I that functions anti-apoptotic. Cer inhibits VEGF. VEGF and IGF-I suppress Cer-induced apoptosis by activating ERK and by inhibiting PKCζ which suppresses Cer-activated SAPK/JNK, respectively. TNF-α-induced Cer production activates PLA2/COX-2, leading to apoptosis. Cer-induced PLA2 activation cleaves AA, and COX-2 is converted into AA, which promotes apoptosis. The binding of Cer to cPLA2 increases AA release, and both cPLA2 and AA increase Cer production, resulting in apoptosis. Cer activates TNF-α-induced NF-κB, cPLA2, AA, LIMK-1 that regulates cytoskeletal organization and IL-1β via inhibition of PKC, leading to apoptosis. Cer inhibits PI3K/Akt/CREB, which functions as anti-apoptotic. Cer suppresses PKC-α that inhibits PP2A activity by decreasing PI3K, which subsequently activates PP2A and results in apoptosis. Cer-induced activation of PP2A inhibits SphK1 activity that functions as anti-apoptotic, leading to apoptosis. Cer increases SGK-1 without increasing its phosphorylation via p38MAPK/cAMP/PKA/PI3K and decreases Akt activity, while overexpression of SGK-1 prevents apoptosis by activating PI3K/Ak. Abbreviations; AA; arachidonic acid, COX-2; cyclooxygenase-2, CREB; cyclic adenosine monophosphate response element-binding protein, ERK; extracellular signal-regulated kinase, IL; interleukin, JNK; Jun N-terminal protein kinase, LIMK-1; LIM kinase-1, MAPK; mitogen-activated protein kinase, MMP-1; matrix metalloproteinase-1, NF-κB; nuclear factor-κB, PI3K; phosphatidylinositol 3-kinase, PKC; protein kinase C, cPLA2; cytosolic phospholipase A2, PP2A; protein phosphatase 2A, SAPK; stress-activated protein kinase, SGK-1; serum- and glucocorticoid-inducible protein kinase-1, SphK1; sphingosine kinase 1, STAT: signal transducer and activator of transcription, TIMP-1; tissue inhibitor of matrix metalloproteinase-1, TNF-α; tumor necrosis factor-α, VEGF; vascular endothelial growth factor. Dashed arrow indicates that Cer increases the expression but no phosphorylation of SGK-1. Red arrow; apoptotic, blue arrow; anti-apoptotic. →; stimulation, dashed arrow, inhibition.

**Figure 6 ijms-23-04010-f006:**
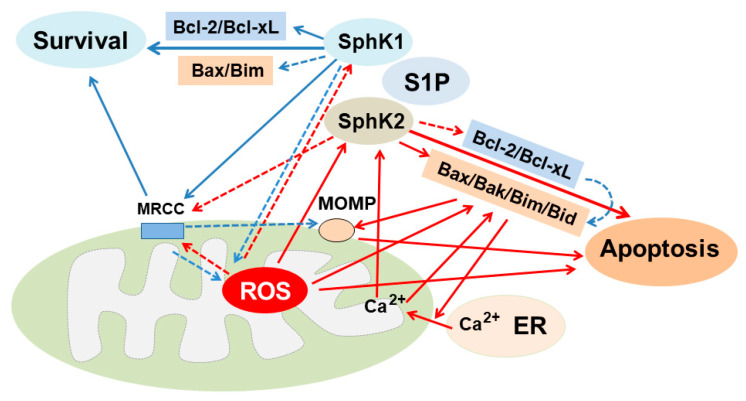
Mitochondria regulate S1P-induced cell survival. SphK1 promotes survival, while SphK2 has the opposite effect. ROS inhibits SphK1/S1P and vice versa. SphK1 prevents ROS-induced apoptosis by preserving MRCC which inhibits MOMP and restores mitochondrial function, leading to a reduction in ROS production, upregulating anti-apoptotic Bcl-2/Bcl-xL and downregulating apoptotic Bax/Bim, while SphK2 has the opposite effect. SphK2 cooperates with C8-BID to stimulate Bax and MOMP, leading to apoptosis, which is inhibited by Bcl-xL. SphK2 activates and inhibits apoptotic Bak and anti-apoptotic Bcl-2 proteins (Bcl-2/Bcl-xL), respectively. SphK2 interacts with Bcl-xL, leading to the suppression of its anti-apoptotic effects. ROS-induced activation of SphK2 induces Ca^2+^ release from the ER, which is dependent on Bax/Bak and dispensable for SphK2 activation, leading to the activation of Bax/Bak and apoptosis. Red arrow; apoptotic pathway, blue arrow; anti-apoptotic pathway, →; stimulation, dashed arrow; inhibition.

**Figure 7 ijms-23-04010-f007:**
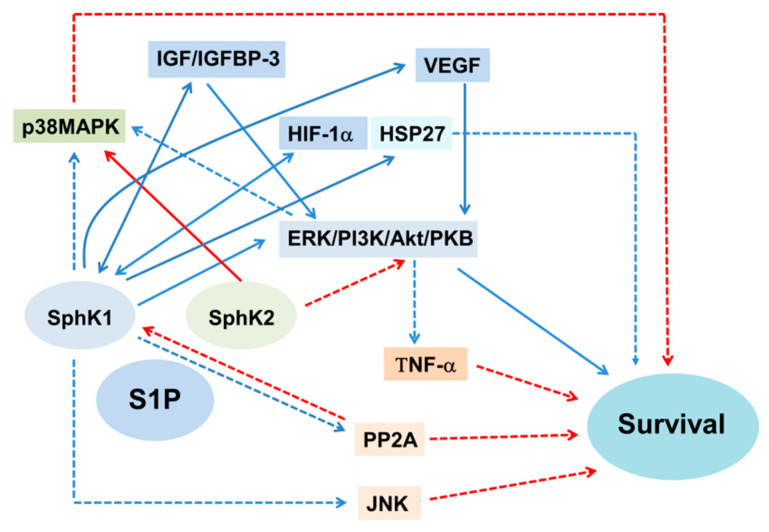
S1P-induced signaling pathways regulate cell survival in oxidant-induced kidney injury. SphK1 prevents ROS-induced apoptosis by activating pro-survival HIF-1α/HSP27. HIF-1α activates SphK1. SphK1 functions as anti-apoptotic by activating ERK/PI3K/Akt/PKB and by inhibiting apoptotic p38MAPK, while SphK2 has the opposite effect. Conversely, ERK activates SphK1. SphK1 activates IGF/IGFBP-3 and vice versa and VEGF, ameliorating ROS-induced apoptosis by activating ERK/PI3K/Akt/PKB. SphK1 inhibits JNK and PP2A, thereby inhibiting apoptosis. Conversely, PP2A inhibits SphK1, leading to apoptosis. Abbreviations; HIF; hypoxia-inducible factor, HSP; heat-shock protein, IGF; insulin growth factor, IGFBP-3; IGF binding protein-3, PI3K; phosphatidylinositol 3-kinase, PKB; protein kinase B, TNF; tumor necrosis factor. Red arrow; apoptotic pathway, blue arrow; anti-apoptotic pathway, →; stimulation, dashed arrow; inhibition.

**Figure 8 ijms-23-04010-f008:**
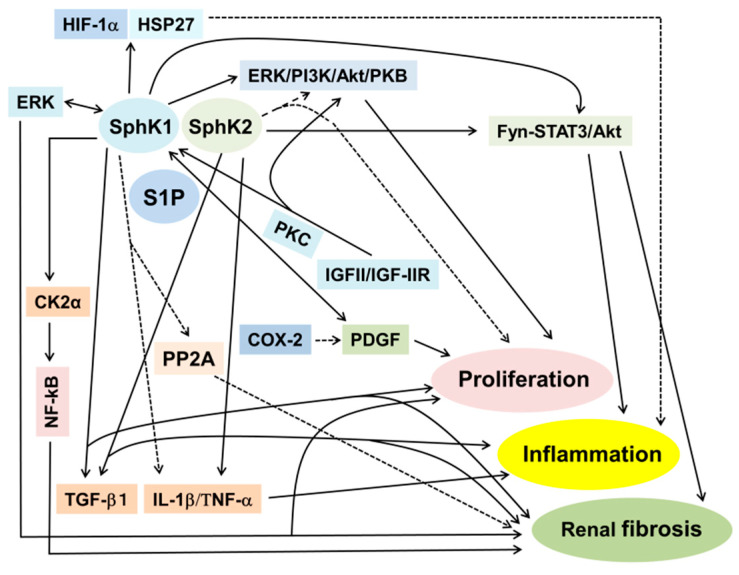
S1P-induced signaling pathways regulate cell proliferation, inflammation and fibrosis in oxidant-induced kidney injury. Cell proliferation: SphK1 promotes cell proliferation via ERK/PI3K/Akt/PKB and TGF-β1, while SphK2 reduces it by inhibiting ERK/PI3K/Akt/PKB. SphK1 activates PDGF by activating ERK, leading to cell proliferation. PDGF promotes cell proliferation by activating SphK1, while this event is inhibited by COX-2. Binding of IGF-II to M6P/IGF-IIR activates ERK through PKC-mediated SphK1 activity, leading to cell proliferation. Inflammation: SphK1/S1P reduces inflammation by inhibiting proinflammatory cytokines (TNF-α and IL-1β) or by activating HIF-1α and HSP27. In contrast, SphK2 enhances inflammation by increasing TGF-β1 expression, inflammatory cytokines (TNF-α and IL-1β). SphK2 together with Fyn promotes inflammation via STAT3/Akt pathway. Renal fibrosis: both SphK1 and SphK2 promote renal fibrosis via the activation of ERK/TGF-β1/CK2α/NF-κB and via Fyn-STAT3/Akt/TGF-β1, respectively. SphK1 also promotes fibrosis by inhibiting PP2A. Abbreviations; CK2α; casein kinase 2α, HIF; hypoxia-inducible factor, HSP; heat-shock protein, IGFII; insulin growth factor II, IGFIIR; IGFII receptor, NF-κB; nuclear factor-κB, PDGF; platelet-derived growth factor, PKB; protein kinase B, PP2A; protein phosphatase 2A, STAT3; signal transducer and activator of transcription 3, TGF; tumor growth factor. →; stimulation, dashed arrow; inhibition.

**Table 1 ijms-23-04010-t001:** Localization of enzymes that regulate formation of ceramide and sphingosine-1-phosphate in kidney.

Enzymes	Localization	References
SPT	kidney, podocyte, RTC, GEC	[14,15]
DES	kidney, RTC	[16,17]
SMS1/2	kidney, RTC, EC	[18,19,20,21]
aSMase	kidney, podocyte, glomerulus, MC, RTC, GEC, EC	[22,23,24,25,26,27,28,29,30,31]
nSMase	kidney, MC, RTC, EC	[22,24,25,28,30,31,33,34]
SMPDL3b	kidney, glomeruli, podocyte, GEC	[13,15,35]
CerK	kidney, podocyte, MC, RTC, GEC	[13,15,36,37]
CerS1	RTC, EC	[38,39,40]
CerS2	kidney, RTC	[40,41]
CerS4	kidney, RTC	[40,41]
CerS5	kidney, RTC	[40,42]
CerS6	kidney, RTC	[40,41]
aCDase	kidney, MC, podocyte, GEC	[24,35,43,44]
nCDase	kidney, glomerulus, podocyte, MC, RTC	[24,25,35,45,46,47]
alkaline CDase	MC, podocyte	[15,35,48]
SphK1	kidney, podocyte, MC, RTC, EC	[17,34,49,51,52,55]
SphK2	kidney, podocyte, MC, RTC, EC	[17,50,51,53,54,55]

aCDase; acid ceramidase, aSMase; acid sphingomyelinase, CerK; ceramide kinase, CerS; ceramide synthase, DES; dihydroceramide desaturase, EC; endothelial cell, GEC; glomerular EC, MC; mesangial cell, nCDase; neutral CDase, nSMase; neutral SMase, RTC; renal tubular cell, SMPDL3b; sphingomyelin phosphodiesterase acid-like 3b, SMS; sphingomyelin synthase, SphKs; sphingosine kinases, SPT; serine palmitoyltransferase.

**Table 2 ijms-23-04010-t002:** Alteration of ceramide levels and kidney cell response in oxidant-induced kidney injury.

Oxidative Kidney Diseases	Cer Levels in Kidney, Plasma/Serum and Urine		Enzymes for Cer Production	Kidney Cell Response	References
Toxic nephropathy	RTC	↑	CerS↑	Apoptosis	[62]
Cadmium					
Carbon tetrachloride	Kidney, plasma	↑	nSMase↑, aSMase↓	Apoptosis/necrosis	[31,63]
Chromium	RTC	↑	SMPD2↑	Autophagy	[64]
Cisplatin	Kidney, BMK	↑	SPT↑, CerS↑, aSMae↑, GCS↑	Apoptosis/necrosis	[65,66]
Nickel	RTC	↑	CerS↑, GCS↑	Apoptosis	[67]
UV-irradiation	RTC, BMK	↑	SPT↑, aSMase↑, CerS↑	Apoptosis	[27,65]
Radiation	Podocyte	↑	SMPDL3b↓, nCDase↓, aCDase→, alkaline CDase→	Apoptosis	[35]
	EC	↑	aSMase↑	Apoptosis	[68,69,70]
	GEC	↑	SMPDL3b↑, CerK→	Apoptosis	[13]
Radiocontrasts	RTC	↑	CerS↑	Apoptosis	[71]
Oxalate nephrolithiasis	RTC	↑	SMase↑	Apoptosis/necrosis	[72,73]
Hypermocysteinemia	Kidney, glomerulus, Podocyte	↑	aSMase↑, SPT↑	MME/necrosis/sclerosis/ fibrosis	[26,74,75]
Myohemoglobinuria	Kidney, RTC	↑	aSMase↓ nSMase↓, CerS↑	Necrosis	[22]
Ischemia/reperfusion	Kidney, RTC	↑	aSMase↓, nSMase↓, CerS↑	Apoptosis/necrosis	[22,76,77,78]
	EC	↑	SMase↑	Necrosis	[79]
Unilateral ureteral obstruction	Kidney	NA	CerS↑	Apoptosis	[71]
	Kidney	↑	UN	Apoptosis	[80,81]
	Kidney	LCCer↓	UN	Fibrosis	[82]
Anti-GBM Ab GN	Kidney	↑	aSMase↑, nSMase↑	Necrosis	[22]
Nephrotic syndrome	Glomerulus, podocyte	↑	aSMase↑, aCDase↓	FPE	[44]
	Podocyte, fibroblast	↑	SGPL1↓, CerS2↑	FPE, FSGS	[83]
	Podocyte, serum	↑	SGPL1↓	FPE, FSGS	[84]
Chronic GN	Plasma/serum	↑	UN	Necrosis	[90,91]
Lupus nephritis	Plasma/serum	↑	UN	Necrosis	[92]
	Plasma	↑	CerS5↑	Necrosis	[93]
	Plasma	→	UN	Necrosis	[94]
Diabetic nephropathy	Kidney	↑	aCDase↓	Apoptosis	[43]
	MC	→	nCDase↑,↓	Proliferation	[47]
	Kidney, RTC	↑	SPT↑, aSMase→	Apoptosis	[85]
	Podocyte	↑	SPT↑	Apoptosis	[86]
	Kidney	↑	UN	NA	[97]
	Kidney	↓	CerS5↓, nSMase↓, alkaline CDase↑	MME	[98]
	EC	↑	aSMase↑, CerS↑	Apoptosis/ necrosis	[87,88,89]
	Plasma	↑	UN	NA	[96]
	Plasma	↑	UN	MME/necrosis	[98,99]
	Plasma	VLCer↓	UN	Necrosis	[101]
	Urine	↑	UN	Necrosis	[102,103]

BMK; baby mouse kidney cell, Cer; ceramide, FPE; foot process effacement, FSGS; focal segmental glomerulosclerosis, GBM Ab; glomerular basement membrane antibody, GCS; glucosylceramide synthase, GN; glomerulonephritis, LCCer; long-chain ceramide, MME; mesangial matrix expansion, SMPD2; sphingomyelin phosphodiesterase-2, SGPL1; S1P lyase, UV; ultraviolet, VLCer; very long-chain Cer. UN; unknown, NA; not available: ↑; increased, ↓; decreased, →; unchanged.

**Table 4 ijms-23-04010-t004:** Overview of compounds that target the enzymes involved in ceramide and S1P metabolism to improve a ceramide-S1P rheostat, contributing to protection against oxidant-induced kidney injury.

Targeted Enzyme	Effector (Function)	Model/Oxidant Stimuli	Tissue/Cell Type/Plasma	Effects	References
SPT	Myriosin (inhibitor)	Cisplatin nephropathy	kidney	Protects against apoptosis/necrosis	[66]
		hHcys	kidney	Protects against proteinuria and necrosis	[74]
		DN	kidney/podocyte	Prevents ROS production, albuminuria and apoptosis	[86]
	Coumesterol (inhibitor)	DM	hepatocyte	Improves insulin resistance	[296]
	NAC * (inhibitor)	obesity/DM	plasma/cardiocyte	Improves insulin resistance	[297]
CerS	Fumonisin B1 (inhibitor)	Cadmium nephropathy	RTC	Prevents apoptosis	[62]
		Radiocontrast nephropathy	RTC	Prevents apoptosis	[71]
		H/R	RTC	Prevents apoptosis/necrosis	[76,78]
CerS4,5	Coumesterol (inhibitor)	DM	hepatocyte	Improves insulin resistance	[296]
CerS5	NAC * (inhibitor)	Obesity/DM	plasma/cardiocyte	Improves insulin resistance	[297]
aSMase	Amitriptyline (inhibitor)	Cisplatin nephropathy	kidney	Prevents apoptosis/necrosis	[66]
		hHcys	podocyte	Protects against ROS production and necrosis	[75]
	Desipramin (inhibitor)	UV-irradiation nephropathy	RTC	Prevents apoptosis	[27]
alSMase, nSMase	NAC* (inhibitor)	Obesity/DM	plasma/cardiocyte	Improves insulin resistance	[297]
aCDase	AdipoRon * (activator)	DN	kidney, podocyte, GEC	Increases S1P and protects against apoptosis	[43,284,298]
nCDase	AdipoRon * (activator)	H_2_O_2_	EC	Improves EC function	[281]
	Dexamethasone ** (activator)	Staurosporine, TNF-α	MC	Prevents apoptosis	[299]
SphK1	SKI-II (inhibitor)	DN	RTC	Prevents inflammation and fibrosis	[200]
	Berberine (inhibitor)	DN	MC	Prevents fibrosis	[198]
			kidney	Prevents injury	[264]
	Dimethylsphingosine (inhibitor)	DN	MC	Prevents fibrosis	[198]
	Dimethylsphingosine (inhibitor)	LDL/DN	MC	Prevents fibrosis	[219]
	Isoflurane (activator)	I/R	kidney	Prevents apoptosis	[285]
	Coumesterol * (activator)	DM	hepatocyte	Improves insulin resistance	[296]
	Dexamethasone (activator)	Staurosporine, TNF-α	MC	Prevents apoptosis	[299]
SphK2	SLM6031434 (inhibitor)	UUO	renal fibroblast	Prevents fibrosis	[286]
	SLM6031434 (inhibitor)	DN	podocyte	Prevents podocyte function and fibrosis	[207]
	HWG-35D (inhibitor)	UUO	renal fibroblast	Prevents fibrosis	[286]
	SLP 120701 (inhibitor)		kidney	Prevents inflammation/fibrosis	[260]
	ABC294640 (inhibitor)	LN	glomeruli	Prevents injury	[94]
	Coumesterol * (activator)	DM	hepatocyte	Improves insulin resistance	[296]
S1PR1	FTY720 (activator)	Cisplatin nephropathy	kidney	Prevents injury	[226]
	FTY720 (activator)	I/R	kidney	Prevents injury	[287]
	SEW2871 (activator)				
	Amiselimod (MT-1303) (activator)	LN	kidney	Prevents injury	[288]
	Ozanimid (RPC-1063) (activator)				[289]
	RP-101075 (activator)				[289]
	KRP-203 (activator)			Prevents proteinuria and apoptosis	[290]
S1PR2	VPC23019 (inhibitor)	IgA GN	MC, RTC	Prevents cell proliferation and fibrosis	[291]
	JTE-013 (inhibitor)	DN	MC	Prevents fibrosis	[292]
			GEC	Prevents ROS production and apoptosis	[293]
	Beberine (inhibitor)		kidney, MC	Prevents fibrosis	[264]
S1PR1/3	VPC23019 (inhibitor)	LDL/DN	MC	Prevents fibrosis	[219]

ACDase; acid ceramidase, AdipoRon; adiponectin receptor agonist, AlSMase; alkaline sphingomyelinase, CerS; ceramide synthase, DM; diabetes mellitus, DN; diabetic nephropathy, EC; endothelial cell, GEC; glomerular endothelial cell, hHcys; hyperhomocysteinemia, H_2_O_2_; hydrogen peroxide, H/R; hypoxia/reperfusion, IgA GN; immunoglobulin A glomerulonephritis, I/R; ischemia/reperfusion, LDL; low-density lipoprotein, LN; lupus nephritis, MC; mesangial cell, NAC; N-acetylcysteine, nCDase; neutral CDase, nSMase; neutral SMase, ROS; reactive oxygen species, RTC; renal tubular cell, aSMase; acid sphingomyelinase, SphK; sphingosine kinase, S1PR; sphingosine-1-phosphate receptor, TNF; tumor necrosis factor, UUO; unilateral ureter obstruction.* These compounds function as an antioxidant. ** Glucocorticoid.

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
