# Peer review of "A Rheostat of Ceramide and Sphingosine-1-Phosphate as a Determinant of Oxidative Stress-Mediated Kidney Injury"

_ijms, 2022, doi:10.3390/ijms23074010_

Round 1

Reviewer 1 Report

The author has adequately addressed all concerns that have been raised in the first report. The manuscript has been significantly improved, and accordingly, I would suggest the publication of this interesting review article.

Author Response

1.This reviewer recommended that “English language and style are fine/minor spell check required”

According to the suggestion, I have carefully checked and revised the manuscript. I have tried to check the revised manuscript as much as I could.

Reviewer 2 Report

Dear Editor,

The manuscript [IJMS] Manuscript ID_ijms-1640020entitled: “A Rheostat of Ceramide and Sphingosine-1-Phosphate as a 2 Determinant of Oxidative Stress-Mediated Kidney Injury” by author Norishi Ueda  represent review article that summarize the current data considering a role of interaction between ROS-antioxidants and ceramide-SphKs/S1P and of a ceramide-S1P rheostat in the regulation of oxidant-induced kidney injury. The review will focus on various types of oxidative stress-mediated kidney diseases, in which increased ROS production and/or
decreased antioxidants can be induced some kidney disease.

It can be seen from the attached manuscript that the author accepted all the suggestions given by other reviewers. That is why the manuscript is now more complete and very "fluent" to read.

Regardless, I also give my opinion on this review work.

General comments:

Title

The title is appropriate, precise and clear for readers.

Abstract

Written clearly and understandably. In brief, Includes all the elements for understanding what is written in the manuscript. Comments of other reviewers are included.

Manuscript

The author in the manuscript provides basic information on the metabolism of ceramides, metabolic pathways of sphingolipids and enzymes involved in these processes. For me, the Interaction between ROS, antioxidants and ceramide in oxidant-induced kidney injury was especially interesting. It is also interesting that decreased glutathione (GSH) and glutathione redox index (GSH/GSSG) enhanced nSMase activity.

Dealing with renal I/R we found that I/R, in most cases, caused an increase in FA concentration (C15:0, C16:1, C17:0, C18:1 c + t, C18:2 c + t, C22:0, and C22:6 + C22:1), with a decrease only in the case of C14:0 and C16:0, and no changes in the case of C18:0. It is known that I/R elevates ROS generation and decreases antioxidant defense through antioxidant enzyme gene downregulation. Studies have reported the loss of the MnSOD enzyme activity following I/R, with the MnSOD inactivation prior to the onset of renal damage, which suggests

an upstream of oxidative damage to renal damage.

We also investigated the protein expression of Nrf2, which has been shown to be the most
important inducible transcription factor that exerts protective effects against renal ischemic damage by regulating the endogenous antioxidant system. Once activated in cytosol, Nrf2 trans-activates antioxidant response elements, which further promote the CuZnSOD, GSH-Px, GST, CAT, and HO-1 expression increase. Several lines of evidence indicate that induced expression of HO-1 attenuates cellular damage and reduces apoptosis, while HO-1 inhibition aggravates it. Our results showed that I/R decreased kidney Nrf2 (16%) and HO-1 (31%) expression levels by 16% and 31%, respectively.

Cellular mechanisms that result in apoptosis represent a cascade of numerous amplifying steps, including activation of pro-apoptotic Bax protein, a member of the Bcl2 protein family. It has been suggested that a delicate interplay between anti- and pro-apoptotic members of the Bcl2 family is necessary for the repair of the damaged renal cells after an ischemic insult. Moreover, it has been reported that the over expression of anti-apoptotic Bcl2 can block both apoptosis and necrosis, as well as protect ischemic tissue against I/R-induced oxidative stress. We also found that meldonium pretreatment reduced Bax/Bcl2 ratio increase for 35%.

I think it would be useful to note this in the context of a possible link between oxidative stress and ceramide metabolism in appropriate places in the text.

Suggestion

This is just one example of how antioxidants can affect the metabolism of ceramides and I think that it will be very usefully to cite following reference together with above mentioned explanations with explanation in places where the author thinks best.

Đurašević S., Stojković M., Bogdanović Lj., Pavlović S., Borković-Mitić S., Grigorov I., Bogojević D., Jasnić N., Tosti T., Đurović S., Đorđević J., Todorović Z. (2019). The effects of meldonium on the renal acute ischemia/reperfusion injury in rats. Int. J. Mol. Sci., 20, 5747; DOI:10.3390/ijms20225747

There are no specific comments in the text.

Conclusion of the Reviewer

The author accepted and in detail explained the suggestions of other reviewers, So, in my opinion, this is a very high quality review manuscript and I suggest to the editor to accept this MS [IJMS] Manuscript ID_ijms-1640020 for publication in this form, but with the proproposed suggestion.

General conclusion: Acceptable for publication.

Dear Editor,

The manuscript [IJMS] Manuscript ID_ijms-1640020entitled: “A Rheostat of Ceramide and Sphingosine-1-Phosphate as a 2 Determinant of Oxidative Stress-Mediated Kidney Injury” by author Norishi Ueda  represent review article that summarize the current data considering a role of interaction between ROS-antioxidants and ceramide-SphKs/S1P and of a ceramide-S1P rheostat in the regulation of oxidant-induced kidney injury. The review will focus on various types of oxidative stress-mediated kidney diseases, in which increased ROS production and/or
decreased antioxidants can be induced some kidney disease.

It can be seen from the attached manuscript that the author accepted all the suggestions given by other reviewers. That is why the manuscript is now more complete and very "fluent" to read.

Regardless, I also give my opinion on this review work.

General comments:

Title

The title is appropriate, precise and clear for readers.

Abstract

Written clearly and understandably. In brief, Includes all the elements for understanding what is written in the manuscript. Comments of other reviewers are included.

Manuscript

The author in the manuscript provides basic information on the metabolism of ceramides, metabolic pathways of sphingolipids and enzymes involved in these processes. For me, the Interaction between ROS, antioxidants and ceramide in oxidant-induced kidney injury was especially interesting. It is also interesting that decreased glutathione (GSH) and glutathione redox index (GSH/GSSG) enhanced nSMase activity.

Dealing with renal I/R we found that I/R, in most cases, caused an increase in FA concentration (C15:0, C16:1, C17:0, C18:1 c + t, C18:2 c + t, C22:0, and C22:6 + C22:1), with a decrease only in the case of C14:0 and C16:0, and no changes in the case of C18:0. It is known that I/R elevates ROS generation and decreases antioxidant defense through antioxidant enzyme gene downregulation. Studies have reported the loss of the MnSOD enzyme activity following I/R, with the MnSOD inactivation prior to the onset of renal damage, which suggests

an upstream of oxidative damage to renal damage.

We also investigated the protein expression of Nrf2, which has been shown to be the most
important inducible transcription factor that exerts protective effects against renal ischemic damage by regulating the endogenous antioxidant system. Once activated in cytosol, Nrf2 trans-activates antioxidant response elements, which further promote the CuZnSOD, GSH-Px, GST, CAT, and HO-1 expression increase. Several lines of evidence indicate that induced expression of HO-1 attenuates cellular damage and reduces apoptosis, while HO-1 inhibition aggravates it. Our results showed that I/R decreased kidney Nrf2 (16%) and HO-1 (31%) expression levels by 16% and 31%, respectively.

Cellular mechanisms that result in apoptosis represent a cascade of numerous amplifying steps, including activation of pro-apoptotic Bax protein, a member of the Bcl2 protein family. It has been suggested that a delicate interplay between anti- and pro-apoptotic members of the Bcl2 family is necessary for the repair of the damaged renal cells after an ischemic insult. Moreover, it has been reported that the over expression of anti-apoptotic Bcl2 can block both apoptosis and necrosis, as well as protect ischemic tissue against I/R-induced oxidative stress. We also found that meldonium pretreatment reduced Bax/Bcl2 ratio increase for 35%.

I think it would be useful to note this in the context of a possible link between oxidative stress and ceramide metabolism in appropriate places in the text.

Suggestion

This is just one example of how antioxidants can affect the metabolism of ceramides and I think that it will be very usefully to cite following reference together with above mentioned explanations with explanation in places where the author thinks best.

Đurašević S., Stojković M., Bogdanović Lj., Pavlović S., Borković-Mitić S., Grigorov I., Bogojević D., Jasnić N., Tosti T., Đurović S., Đorđević J., Todorović Z. (2019). The effects of meldonium on the renal acute ischemia/reperfusion injury in rats. Int. J. Mol. Sci., 20, 5747; DOI:10.3390/ijms20225747

There are no specific comments in the text.

Conclusion of the Reviewer

The author accepted and in detail explained the suggestions of other reviewers, So, in my opinion, this is a very high quality review manuscript and I suggest to the editor to accept this MS [IJMS] Manuscript ID_ijms-1640020 for publication in this form, but with the proproposed suggestion.

General conclusion: Acceptable for publication.

Dear Editor,

The manuscript [IJMS] Manuscript ID_ijms-1640020entitled: “A Rheostat of Ceramide and Sphingosine-1-Phosphate as a 2 Determinant of Oxidative Stress-Mediated Kidney Injury” by author Norishi Ueda  represent review article that summarize the current data considering a role of interaction between ROS-antioxidants and ceramide-SphKs/S1P and of a ceramide-S1P rheostat in the regulation of oxidant-induced kidney injury. The review will focus on various types of oxidative stress-mediated kidney diseases, in which increased ROS production and/or
decreased antioxidants can be induced some kidney disease.

It can be seen from the attached manuscript that the author accepted all the suggestions given by other reviewers. That is why the manuscript is now more complete and very "fluent" to read.

Regardless, I also give my opinion on this review work.

General comments:

Title

The title is appropriate, precise and clear for readers.

Abstract

Written clearly and understandably. In brief, Includes all the elements for understanding what is written in the manuscript. Comments of other reviewers are included.

Manuscript

The author in the manuscript provides basic information on the metabolism of ceramides, metabolic pathways of sphingolipids and enzymes involved in these processes. For me, the Interaction between ROS, antioxidants and ceramide in oxidant-induced kidney injury was especially interesting. It is also interesting that decreased glutathione (GSH) and glutathione redox index (GSH/GSSG) enhanced nSMase activity.

Dealing with renal I/R we found that I/R, in most cases, caused an increase in FA concentration (C15:0, C16:1, C17:0, C18:1 c + t, C18:2 c + t, C22:0, and C22:6 + C22:1), with a decrease only in the case of C14:0 and C16:0, and no changes in the case of C18:0. It is known that I/R elevates ROS generation and decreases antioxidant defense through antioxidant enzyme gene downregulation. Studies have reported the loss of the MnSOD enzyme activity following I/R, with the MnSOD inactivation prior to the onset of renal damage, which suggests

an upstream of oxidative damage to renal damage.

We also investigated the protein expression of Nrf2, which has been shown to be the most
important inducible transcription factor that exerts protective effects against renal ischemic damage by regulating the endogenous antioxidant system. Once activated in cytosol, Nrf2 trans-activates antioxidant response elements, which further promote the CuZnSOD, GSH-Px, GST, CAT, and HO-1 expression increase. Several lines of evidence indicate that induced expression of HO-1 attenuates cellular damage and reduces apoptosis, while HO-1 inhibition aggravates it. Our results showed that I/R decreased kidney Nrf2 (16%) and HO-1 (31%) expression levels by 16% and 31%, respectively.

Cellular mechanisms that result in apoptosis represent a cascade of numerous amplifying steps, including activation of pro-apoptotic Bax protein, a member of the Bcl2 protein family. It has been suggested that a delicate interplay between anti- and pro-apoptotic members of the Bcl2 family is necessary for the repair of the damaged renal cells after an ischemic insult. Moreover, it has been reported that the over expression of anti-apoptotic Bcl2 can block both apoptosis and necrosis, as well as protect ischemic tissue against I/R-induced oxidative stress. We also found that meldonium pretreatment reduced Bax/Bcl2 ratio increase for 35%.

I think it would be useful to note this in the context of a possible link between oxidative stress and ceramide metabolism in appropriate places in the text.

Suggestion

This is just one example of how antioxidants can affect the metabolism of ceramides and I think that it will be very usefully to cite following reference together with above mentioned explanations with explanation in places where the author thinks best.

Đurašević S., Stojković M., Bogdanović Lj., Pavlović S., Borković-Mitić S., Grigorov I., Bogojević D., Jasnić N., Tosti T., Đurović S., Đorđević J., Todorović Z. (2019). The effects of meldonium on the renal acute ischemia/reperfusion injury in rats. Int. J. Mol. Sci., 20, 5747; DOI:10.3390/ijms20225747

There are no specific comments in the text.

Conclusion of the Reviewer

The author accepted and in detail explained the suggestions of other reviewers, So, in my opinion, this is a very high quality review manuscript and I suggest to the editor to accept this MS [IJMS] Manuscript ID_ijms-1640020 for publication in this form, but with the proproposed suggestion.

General conclusion: Acceptable for publication.

Dear Editor,

The manuscript [IJMS] Manuscript ID_ijms-1640020entitled: “A Rheostat of Ceramide and Sphingosine-1-Phosphate as a 2 Determinant of Oxidative Stress-Mediated Kidney Injury” by author Norishi Ueda  represent review article that summarize the current data considering a role of interaction between ROS-antioxidants and ceramide-SphKs/S1P and of a ceramide-S1P rheostat in the regulation of oxidant-induced kidney injury. The review will focus on various types of oxidative stress-mediated kidney diseases, in which increased ROS production and/or
decreased antioxidants can be induced some kidney disease.

It can be seen from the attached manuscript that the author accepted all the suggestions given by other reviewers. That is why the manuscript is now more complete and very "fluent" to read.

Regardless, I also give my opinion on this review work.

General comments:

Title

The title is appropriate, precise and clear for readers.

Abstract

Written clearly and understandably. In brief, Includes all the elements for understanding what is written in the manuscript. Comments of other reviewers are included.

Manuscript

The author in the manuscript provides basic information on the metabolism of ceramides, metabolic pathways of sphingolipids and enzymes involved in these processes. For me, the Interaction between ROS, antioxidants and ceramide in oxidant-induced kidney injury was especially interesting. It is also interesting that decreased glutathione (GSH) and glutathione redox index (GSH/GSSG) enhanced nSMase activity.

Dealing with renal I/R we found that I/R, in most cases, caused an increase in FA concentration (C15:0, C16:1, C17:0, C18:1 c + t, C18:2 c + t, C22:0, and C22:6 + C22:1), with a decrease only in the case of C14:0 and C16:0, and no changes in the case of C18:0. It is known that I/R elevates ROS generation and decreases antioxidant defense through antioxidant enzyme gene downregulation. Studies have reported the loss of the MnSOD enzyme activity following I/R, with the MnSOD inactivation prior to the onset of renal damage, which suggests

an upstream of oxidative damage to renal damage.

We also investigated the protein expression of Nrf2, which has been shown to be the most
important inducible transcription factor that exerts protective effects against renal ischemic damage by regulating the endogenous antioxidant system. Once activated in cytosol, Nrf2 trans-activates antioxidant response elements, which further promote the CuZnSOD, GSH-Px, GST, CAT, and HO-1 expression increase. Several lines of evidence indicate that induced expression of HO-1 attenuates cellular damage and reduces apoptosis, while HO-1 inhibition aggravates it. Our results showed that I/R decreased kidney Nrf2 (16%) and HO-1 (31%) expression levels by 16% and 31%, respectively.

Cellular mechanisms that result in apoptosis represent a cascade of numerous amplifying steps, including activation of pro-apoptotic Bax protein, a member of the Bcl2 protein family. It has been suggested that a delicate interplay between anti- and pro-apoptotic members of the Bcl2 family is necessary for the repair of the damaged renal cells after an ischemic insult. Moreover, it has been reported that the over expression of anti-apoptotic Bcl2 can block both apoptosis and necrosis, as well as protect ischemic tissue against I/R-induced oxidative stress. We also found that meldonium pretreatment reduced Bax/Bcl2 ratio increase for 35%.

I think it would be useful to note this in the context of a possible link between oxidative stress and ceramide metabolism in appropriate places in the text.

Suggestion

This is just one example of how antioxidants can affect the metabolism of ceramides and I think that it will be very usefully to cite following reference together with above mentioned explanations with explanation in places where the author thinks best.

Đurašević S., Stojković M., Bogdanović Lj., Pavlović S., Borković-Mitić S., Grigorov I., Bogojević D., Jasnić N., Tosti T., Đurović S., Đorđević J., Todorović Z. (2019). The effects of meldonium on the renal acute ischemia/reperfusion injury in rats. Int. J. Mol. Sci., 20, 5747; DOI:10.3390/ijms20225747

There are no specific comments in the text.

Conclusion of the Reviewer

The author accepted and in detail explained the suggestions of other reviewers, So, in my opinion, this is a very high quality review manuscript and I suggest to the editor to accept this MS [IJMS] Manuscript ID_ijms-1640020 for publication in this form, but with the proproposed suggestion.

General conclusion: Acceptable for publication.

Dear Editor,

The manuscript [IJMS] Manuscript ID_ijms-1640020entitled: “A Rheostat of Ceramide and Sphingosine-1-Phosphate as a 2 Determinant of Oxidative Stress-Mediated Kidney Injury” by author Norishi Ueda  represent review article that summarize the current data considering a role of interaction between ROS-antioxidants and ceramide-SphKs/S1P and of a ceramide-S1P rheostat in the regulation of oxidant-induced kidney injury. The review will focus on various types of oxidative stress-mediated kidney diseases, in which increased ROS production and/or
decreased antioxidants can be induced some kidney disease.

It can be seen from the attached manuscript that the author accepted all the suggestions given by other reviewers. That is why the manuscript is now more complete and very "fluent" to read.

Regardless, I also give my opinion on this review work.

General comments:

Title

The title is appropriate, precise and clear for readers.

Abstract

Written clearly and understandably. In brief, Includes all the elements for understanding what is written in the manuscript. Comments of other reviewers are included.

Manuscript

The author in the manuscript provides basic information on the metabolism of ceramides, metabolic pathways of sphingolipids and enzymes involved in these processes. For me, the Interaction between ROS, antioxidants and ceramide in oxidant-induced kidney injury was especially interesting. It is also interesting that decreased glutathione (GSH) and glutathione redox index (GSH/GSSG) enhanced nSMase activity.

Dealing with renal I/R we found that I/R, in most cases, caused an increase in FA concentration (C15:0, C16:1, C17:0, C18:1 c + t, C18:2 c + t, C22:0, and C22:6 + C22:1), with a decrease only in the case of C14:0 and C16:0, and no changes in the case of C18:0. It is known that I/R elevates ROS generation and decreases antioxidant defense through antioxidant enzyme gene downregulation. Studies have reported the loss of the MnSOD enzyme activity following I/R, with the MnSOD inactivation prior to the onset of renal damage, which suggests

an upstream of oxidative damage to renal damage.

We also investigated the protein expression of Nrf2, which has been shown to be the most
important inducible transcription factor that exerts protective effects against renal ischemic damage by regulating the endogenous antioxidant system. Once activated in cytosol, Nrf2 trans-activates antioxidant response elements, which further promote the CuZnSOD, GSH-Px, GST, CAT, and HO-1 expression increase. Several lines of evidence indicate that induced expression of HO-1 attenuates cellular damage and reduces apoptosis, while HO-1 inhibition aggravates it. Our results showed that I/R decreased kidney Nrf2 (16%) and HO-1 (31%) expression levels by 16% and 31%, respectively.

Cellular mechanisms that result in apoptosis represent a cascade of numerous amplifying steps, including activation of pro-apoptotic Bax protein, a member of the Bcl2 protein family. It has been suggested that a delicate interplay between anti- and pro-apoptotic members of the Bcl2 family is necessary for the repair of the damaged renal cells after an ischemic insult. Moreover, it has been reported that the over expression of anti-apoptotic Bcl2 can block both apoptosis and necrosis, as well as protect ischemic tissue against I/R-induced oxidative stress. We also found that meldonium pretreatment reduced Bax/Bcl2 ratio increase for 35%.

I think it would be useful to note this in the context of a possible link between oxidative stress and ceramide metabolism in appropriate places in the text.

Suggestion

This is just one example of how antioxidants can affect the metabolism of ceramides and I think that it will be very usefully to cite following reference together with above mentioned explanations with explanation in places where the author thinks best.

Đurašević S., Stojković M., Bogdanović Lj., Pavlović S., Borković-Mitić S., Grigorov I., Bogojević D., Jasnić N., Tosti T., Đurović S., Đorđević J., Todorović Z. (2019). The effects of meldonium on the renal acute ischemia/reperfusion injury in rats. Int. J. Mol. Sci., 20, 5747; DOI:10.3390/ijms20225747

There are no specific comments in the text.

Conclusion of the Reviewer

The author accepted and in detail explained the suggestions of other reviewers, So, in my opinion, this is a very high quality review manuscript and I suggest to the editor to accept this MS [IJMS] Manuscript ID_ijms-1640020 for publication in this form, but with the proproposed suggestion.

General conclusion: Acceptable for publication.

Dear Editor,

The manuscript [IJMS] Manuscript ID_ijms-1640020entitled: “A Rheostat of Ceramide and Sphingosine-1-Phosphate as a 2 Determinant of Oxidative Stress-Mediated Kidney Injury” by author Norishi Ueda  represent review article that summarize the current data considering a role of interaction between ROS-antioxidants and ceramide-SphKs/S1P and of a ceramide-S1P rheostat in the regulation of oxidant-induced kidney injury. The review will focus on various types of oxidative stress-mediated kidney diseases, in which increased ROS production and/or
decreased antioxidants can be induced some kidney disease.

It can be seen from the attached manuscript that the author accepted all the suggestions given by other reviewers. That is why the manuscript is now more complete and very "fluent" to read.

Regardless, I also give my opinion on this review work.

General comments:

Title

The title is appropriate, precise and clear for readers.

Abstract

Written clearly and understandably. In brief, Includes all the elements for understanding what is written in the manuscript. Comments of other reviewers are included.

Manuscript

The author in the manuscript provides basic information on the metabolism of ceramides, metabolic pathways of sphingolipids and enzymes involved in these processes. For me, the Interaction between ROS, antioxidants and ceramide in oxidant-induced kidney injury was especially interesting. It is also interesting that decreased glutathione (GSH) and glutathione redox index (GSH/GSSG) enhanced nSMase activity.

Dealing with renal I/R we found that I/R, in most cases, caused an increase in FA concentration (C15:0, C16:1, C17:0, C18:1 c + t, C18:2 c + t, C22:0, and C22:6 + C22:1), with a decrease only in the case of C14:0 and C16:0, and no changes in the case of C18:0. It is known that I/R elevates ROS generation and decreases antioxidant defense through antioxidant enzyme gene downregulation. Studies have reported the loss of the MnSOD enzyme activity following I/R, with the MnSOD inactivation prior to the onset of renal damage, which suggests

an upstream of oxidative damage to renal damage.

We also investigated the protein expression of Nrf2, which has been shown to be the most
important inducible transcription factor that exerts protective effects against renal ischemic damage by regulating the endogenous antioxidant system. Once activated in cytosol, Nrf2 trans-activates antioxidant response elements, which further promote the CuZnSOD, GSH-Px, GST, CAT, and HO-1 expression increase. Several lines of evidence indicate that induced expression of HO-1 attenuates cellular damage and reduces apoptosis, while HO-1 inhibition aggravates it. Our results showed that I/R decreased kidney Nrf2 (16%) and HO-1 (31%) expression levels by 16% and 31%, respectively.

Cellular mechanisms that result in apoptosis represent a cascade of numerous amplifying steps, including activation of pro-apoptotic Bax protein, a member of the Bcl2 protein family. It has been suggested that a delicate interplay between anti- and pro-apoptotic members of the Bcl2 family is necessary for the repair of the damaged renal cells after an ischemic insult. Moreover, it has been reported that the over expression of anti-apoptotic Bcl2 can block both apoptosis and necrosis, as well as protect ischemic tissue against I/R-induced oxidative stress. We also found that meldonium pretreatment reduced Bax/Bcl2 ratio increase for 35%.

I think it would be useful to note this in the context of a possible link between oxidative stress and ceramide metabolism in appropriate places in the text.

Suggestion

This is just one example of how antioxidants can affect the metabolism of ceramides and I think that it will be very usefully to cite following reference together with above mentioned explanations with explanation in places where the author thinks best.

Đurašević S., Stojković M., Bogdanović Lj., Pavlović S., Borković-Mitić S., Grigorov I., Bogojević D., Jasnić N., Tosti T., Đurović S., Đorđević J., Todorović Z. (2019). The effects of meldonium on the renal acute ischemia/reperfusion injury in rats. Int. J. Mol. Sci., 20, 5747; DOI:10.3390/ijms20225747

There are no specific comments in the text.

Conclusion of the Reviewer

The author accepted and in detail explained the suggestions of other reviewers, So, in my opinion, this is a very high quality review manuscript and I suggest to the editor to accept this MS [IJMS] Manuscript ID_ijms-1640020 for publication in this form, but with the proproposed suggestion.

General conclusion: Acceptable for publication.

Dear Editor,

The manuscript [IJMS] Manuscript ID_ijms-1640020entitled: “A Rheostat of Ceramide and Sphingosine-1-Phosphate as a 2 Determinant of Oxidative Stress-Mediated Kidney Injury” by author Norishi Ueda  represent review article that summarize the current data considering a role of interaction between ROS-antioxidants and ceramide-SphKs/S1P and of a ceramide-S1P rheostat in the regulation of oxidant-induced kidney injury. The review will focus on various types of oxidative stress-mediated kidney diseases, in which increased ROS production and/or
decreased antioxidants can be induced some kidney disease.

It can be seen from the attached manuscript that the author accepted all the suggestions given by other reviewers. That is why the manuscript is now more complete and very "fluent" to read.

Regardless, I also give my opinion on this review work.

General comments:

Title

The title is appropriate, precise and clear for readers.

Abstract

Written clearly and understandably. In brief, Includes all the elements for understanding what is written in the manuscript. Comments of other reviewers are included.

Manuscript

The author in the manuscript provides basic information on the metabolism of ceramides, metabolic pathways of sphingolipids and enzymes involved in these processes. For me, the Interaction between ROS, antioxidants and ceramide in oxidant-induced kidney injury was especially interesting. It is also interesting that decreased glutathione (GSH) and glutathione redox index (GSH/GSSG) enhanced nSMase activity.

Dealing with renal I/R we found that I/R, in most cases, caused an increase in FA concentration (C15:0, C16:1, C17:0, C18:1 c + t, C18:2 c + t, C22:0, and C22:6 + C22:1), with a decrease only in the case of C14:0 and C16:0, and no changes in the case of C18:0. It is known that I/R elevates ROS generation and decreases antioxidant defense through antioxidant enzyme gene downregulation. Studies have reported the loss of the MnSOD enzyme activity following I/R, with the MnSOD inactivation prior to the onset of renal damage, which suggests

an upstream of oxidative damage to renal damage.

We also investigated the protein expression of Nrf2, which has been shown to be the most
important inducible transcription factor that exerts protective effects against renal ischemic damage by regulating the endogenous antioxidant system. Once activated in cytosol, Nrf2 trans-activates antioxidant response elements, which further promote the CuZnSOD, GSH-Px, GST, CAT, and HO-1 expression increase. Several lines of evidence indicate that induced expression of HO-1 attenuates cellular damage and reduces apoptosis, while HO-1 inhibition aggravates it. Our results showed that I/R decreased kidney Nrf2 (16%) and HO-1 (31%) expression levels by 16% and 31%, respectively.

Cellular mechanisms that result in apoptosis represent a cascade of numerous amplifying steps, including activation of pro-apoptotic Bax protein, a member of the Bcl2 protein family. It has been suggested that a delicate interplay between anti- and pro-apoptotic members of the Bcl2 family is necessary for the repair of the damaged renal cells after an ischemic insult. Moreover, it has been reported that the over expression of anti-apoptotic Bcl2 can block both apoptosis and necrosis, as well as protect ischemic tissue against I/R-induced oxidative stress. We also found that meldonium pretreatment reduced Bax/Bcl2 ratio increase for 35%.

I think it would be useful to note this in the context of a possible link between oxidative stress and ceramide metabolism in appropriate places in the text.

Suggestion

This is just one example of how antioxidants can affect the metabolism of ceramides and I think that it will be very usefully to cite following reference together with above mentioned explanations with explanation in places where the author thinks best.

Đurašević S., Stojković M., Bogdanović Lj., Pavlović S., Borković-Mitić S., Grigorov I., Bogojević D., Jasnić N., Tosti T., Đurović S., Đorđević J., Todorović Z. (2019). The effects of meldonium on the renal acute ischemia/reperfusion injury in rats. Int. J. Mol. Sci., 20, 5747; DOI:10.3390/ijms20225747

There are no specific comments in the text.

Conclusion of the Reviewer

The author accepted and in detail explained the suggestions of other reviewers, So, in my opinion, this is a very high quality review manuscript and I suggest to the editor to accept this MS [IJMS] Manuscript ID_ijms-1640020 for publication in this form, but with the proproposed suggestion.

General conclusion: Acceptable for publication.

Dear Editor,

The manuscript [IJMS] Manuscript ID_ijms-1640020entitled: “A Rheostat of Ceramide and Sphingosine-1-Phosphate as a 2 Determinant of Oxidative Stress-Mediated Kidney Injury” by author Norishi Ueda  represent review article that summarize the current data considering a role of interaction between ROS-antioxidants and ceramide-SphKs/S1P and of a ceramide-S1P rheostat in the regulation of oxidant-induced kidney injury. The review will focus on various types of oxidative stress-mediated kidney diseases, in which increased ROS production and/or
decreased antioxidants can be induced some kidney disease.

It can be seen from the attached manuscript that the author accepted all the suggestions given by other reviewers. That is why the manuscript is now more complete and very "fluent" to read.

Regardless, I also give my opinion on this review work.

General comments:

Title

The title is appropriate, precise and clear for readers.

Abstract

Written clearly and understandably. In brief, Includes all the elements for understanding what is written in the manuscript. Comments of other reviewers are included.

Manuscript

The author in the manuscript provides basic information on the metabolism of ceramides, metabolic pathways of sphingolipids and enzymes involved in these processes. For me, the Interaction between ROS, antioxidants and ceramide in oxidant-induced kidney injury was especially interesting. It is also interesting that decreased glutathione (GSH) and glutathione redox index (GSH/GSSG) enhanced nSMase activity.

Dealing with renal I/R we found that I/R, in most cases, caused an increase in FA concentration (C15:0, C16:1, C17:0, C18:1 c + t, C18:2 c + t, C22:0, and C22:6 + C22:1), with a decrease only in the case of C14:0 and C16:0, and no changes in the case of C18:0. It is known that I/R elevates ROS generation and decreases antioxidant defense through antioxidant enzyme gene downregulation. Studies have reported the loss of the MnSOD enzyme activity following I/R, with the MnSOD inactivation prior to the onset of renal damage, which suggests

an upstream of oxidative damage to renal damage.

We also investigated the protein expression of Nrf2, which has been shown to be the most
important inducible transcription factor that exerts protective effects against renal ischemic damage by regulating the endogenous antioxidant system. Once activated in cytosol, Nrf2 trans-activates antioxidant response elements, which further promote the CuZnSOD, GSH-Px, GST, CAT, and HO-1 expression increase. Several lines of evidence indicate that induced expression of HO-1 attenuates cellular damage and reduces apoptosis, while HO-1 inhibition aggravates it. Our results showed that I/R decreased kidney Nrf2 (16%) and HO-1 (31%) expression levels by 16% and 31%, respectively.

Cellular mechanisms that result in apoptosis represent a cascade of numerous amplifying steps, including activation of pro-apoptotic Bax protein, a member of the Bcl2 protein family. It has been suggested that a delicate interplay between anti- and pro-apoptotic members of the Bcl2 family is necessary for the repair of the damaged renal cells after an ischemic insult. Moreover, it has been reported that the over expression of anti-apoptotic Bcl2 can block both apoptosis and necrosis, as well as protect ischemic tissue against I/R-induced oxidative stress. We also found that meldonium pretreatment reduced Bax/Bcl2 ratio increase for 35%.

I think it would be useful to note this in the context of a possible link between oxidative stress and ceramide metabolism in appropriate places in the text.

Suggestion

This is just one example of how antioxidants can affect the metabolism of ceramides and I think that it will be very usefully to cite following reference together with above mentioned explanations with explanation in places where the author thinks best.

Đurašević S., Stojković M., Bogdanović Lj., Pavlović S., Borković-Mitić S., Grigorov I., Bogojević D., Jasnić N., Tosti T., Đurović S., Đorđević J., Todorović Z. (2019). The effects of meldonium on the renal acute ischemia/reperfusion injury in rats. Int. J. Mol. Sci., 20, 5747; DOI:10.3390/ijms20225747

There are no specific comments in the text.

Conclusion of the Reviewer

The author accepted and in detail explained the suggestions of other reviewers, So, in my opinion, this is a very high quality review manuscript and I suggest to the editor to accept this MS [IJMS] Manuscript ID_ijms-1640020 for publication in this form, but with the proproposed suggestion.

General conclusion: Acceptable for publication.

Dear Editor,

The manuscript [IJMS] Manuscript ID_ijms-1640020entitled: “A Rheostat of Ceramide and Sphingosine-1-Phosphate as a 2 Determinant of Oxidative Stress-Mediated Kidney Injury” by author Norishi Ueda  represent review article that summarize the current data considering a role of interaction between ROS-antioxidants and ceramide-SphKs/S1P and of a ceramide-S1P rheostat in the regulation of oxidant-induced kidney injury. The review will focus on various types of oxidative stress-mediated kidney diseases, in which increased ROS production and/or
decreased antioxidants can be induced some kidney disease.

It can be seen from the attached manuscript that the author accepted all the suggestions given by other reviewers. That is why the manuscript is now more complete and very "fluent" to read.

Regardless, I also give my opinion on this review work.

General comments:

Title

The title is appropriate, precise and clear for readers.

Abstract

Written clearly and understandably. In brief, Includes all the elements for understanding what is written in the manuscript. Comments of other reviewers are included.

Manuscript

The author in the manuscript provides basic information on the metabolism of ceramides, metabolic pathways of sphingolipids and enzymes involved in these processes. For me, the Interaction between ROS, antioxidants and ceramide in oxidant-induced kidney injury was especially interesting. It is also interesting that decreased glutathione (GSH) and glutathione redox index (GSH/GSSG) enhanced nSMase activity.

Dealing with renal I/R we found that I/R, in most cases, caused an increase in FA concentration (C15:0, C16:1, C17:0, C18:1 c + t, C18:2 c + t, C22:0, and C22:6 + C22:1), with a decrease only in the case of C14:0 and C16:0, and no changes in the case of C18:0. It is known that I/R elevates ROS generation and decreases antioxidant defense through antioxidant enzyme gene downregulation. Studies have reported the loss of the MnSOD enzyme activity following I/R, with the MnSOD inactivation prior to the onset of renal damage, which suggests

an upstream of oxidative damage to renal damage.

We also investigated the protein expression of Nrf2, which has been shown to be the most
important inducible transcription factor that exerts protective effects against renal ischemic damage by regulating the endogenous antioxidant system. Once activated in cytosol, Nrf2 trans-activates antioxidant response elements, which further promote the CuZnSOD, GSH-Px, GST, CAT, and HO-1 expression increase. Several lines of evidence indicate that induced expression of HO-1 attenuates cellular damage and reduces apoptosis, while HO-1 inhibition aggravates it. Our results showed that I/R decreased kidney Nrf2 (16%) and HO-1 (31%) expression levels by 16% and 31%, respectively.

Cellular mechanisms that result in apoptosis represent a cascade of numerous amplifying steps, including activation of pro-apoptotic Bax protein, a member of the Bcl2 protein family. It has been suggested that a delicate interplay between anti- and pro-apoptotic members of the Bcl2 family is necessary for the repair of the damaged renal cells after an ischemic insult. Moreover, it has been reported that the over expression of anti-apoptotic Bcl2 can block both apoptosis and necrosis, as well as protect ischemic tissue against I/R-induced oxidative stress. We also found that meldonium pretreatment reduced Bax/Bcl2 ratio increase for 35%.

I think it would be useful to note this in the context of a possible link between oxidative stress and ceramide metabolism in appropriate places in the text.

Suggestion

This is just one example of how antioxidants can affect the metabolism of ceramides and I think that it will be very usefully to cite following reference together with above mentioned explanations with explanation in places where the author thinks best.

Đurašević S., Stojković M., Bogdanović Lj., Pavlović S., Borković-Mitić S., Grigorov I., Bogojević D., Jasnić N., Tosti T., Đurović S., Đorđević J., Todorović Z. (2019). The effects of meldonium on the renal acute ischemia/reperfusion injury in rats. Int. J. Mol. Sci., 20, 5747; DOI:10.3390/ijms20225747

There are no specific comments in the text.

Conclusion of the Reviewer

The author accepted and in detail explained the suggestions of other reviewers, So, in my opinion, this is a very high quality review manuscript and I suggest to the editor to accept this MS [IJMS] Manuscript ID_ijms-1640020 for publication in this form, but with the proproposed suggestion.

General conclusion: Acceptable for publication.

Dear Editor,

The manuscript [IJMS] Manuscript ID_ijms-1640020entitled: “A Rheostat of Ceramide and Sphingosine-1-Phosphate as a 2 Determinant of Oxidative Stress-Mediated Kidney Injury” by author Norishi Ueda  represent review article that summarize the current data considering a role of interaction between ROS-antioxidants and ceramide-SphKs/S1P and of a ceramide-S1P rheostat in the regulation of oxidant-induced kidney injury. The review will focus on various types of oxidative stress-mediated kidney diseases, in which increased ROS production and/or
decreased antioxidants can be induced some kidney disease.

It can be seen from the attached manuscript that the author accepted all the suggestions given by other reviewers. That is why the manuscript is now more complete and very "fluent" to read.

Regardless, I also give my opinion on this review work.

General comments:

Title

The title is appropriate, precise and clear for readers.

Abstract

Written clearly and understandably. In brief, Includes all the elements for understanding what is written in the manuscript. Comments of other reviewers are included.

Manuscript

The author in the manuscript provides basic information on the metabolism of ceramides, metabolic pathways of sphingolipids and enzymes involved in these processes. For me, the Interaction between ROS, antioxidants and ceramide in oxidant-induced kidney injury was especially interesting. It is also interesting that decreased glutathione (GSH) and glutathione redox index (GSH/GSSG) enhanced nSMase activity.

Dealing with renal I/R we found that I/R, in most cases, caused an increase in FA concentration (C15:0, C16:1, C17:0, C18:1 c + t, C18:2 c + t, C22:0, and C22:6 + C22:1), with a decrease only in the case of C14:0 and C16:0, and no changes in the case of C18:0. It is known that I/R elevates ROS generation and decreases antioxidant defense through antioxidant enzyme gene downregulation. Studies have reported the loss of the MnSOD enzyme activity following I/R, with the MnSOD inactivation prior to the onset of renal damage, which suggests

an upstream of oxidative damage to renal damage.

We also investigated the protein expression of Nrf2, which has been shown to be the most
important inducible transcription factor that exerts protective effects against renal ischemic damage by regulating the endogenous antioxidant system. Once activated in cytosol, Nrf2 trans-activates antioxidant response elements, which further promote the CuZnSOD, GSH-Px, GST, CAT, and HO-1 expression increase. Several lines of evidence indicate that induced expression of HO-1 attenuates cellular damage and reduces apoptosis, while HO-1 inhibition aggravates it. Our results showed that I/R decreased kidney Nrf2 (16%) and HO-1 (31%) expression levels by 16% and 31%, respectively.

Cellular mechanisms that result in apoptosis represent a cascade of numerous amplifying steps, including activation of pro-apoptotic Bax protein, a member of the Bcl2 protein family. It has been suggested that a delicate interplay between anti- and pro-apoptotic members of the Bcl2 family is necessary for the repair of the damaged renal cells after an ischemic insult. Moreover, it has been reported that the over expression of anti-apoptotic Bcl2 can block both apoptosis and necrosis, as well as protect ischemic tissue against I/R-induced oxidative stress. We also found that meldonium pretreatment reduced Bax/Bcl2 ratio increase for 35%.

I think it would be useful to note this in the context of a possible link between oxidative stress and ceramide metabolism in appropriate places in the text.

Suggestion

This is just one example of how antioxidants can affect the metabolism of ceramides and I think that it will be very usefully to cite following reference together with above mentioned explanations with explanation in places where the author thinks best.

Đurašević S., Stojković M., Bogdanović Lj., Pavlović S., Borković-Mitić S., Grigorov I., Bogojević D., Jasnić N., Tosti T., Đurović S., Đorđević J., Todorović Z. (2019). The effects of meldonium on the renal acute ischemia/reperfusion injury in rats. Int. J. Mol. Sci., 20, 5747; DOI:10.3390/ijms20225747

There are no specific comments in the text.

Conclusion of the Reviewer

The author accepted and in detail explained the suggestions of other reviewers, So, in my opinion, this is a very high quality review manuscript and I suggest to the editor to accept this MS [IJMS] Manuscript ID_ijms-1640020 for publication in this form, but with the proproposed suggestion.

General conclusion: Acceptable for publication.

Dear Editor,

The manuscript [IJMS] Manuscript ID_ijms-1640020entitled: “A Rheostat of Ceramide and Sphingosine-1-Phosphate as a 2 Determinant of Oxidative Stress-Mediated Kidney Injury” by author Norishi Ueda  represent review article that summarize the current data considering a role of interaction between ROS-antioxidants and ceramide-SphKs/S1P and of a ceramide-S1P rheostat in the regulation of oxidant-induced kidney injury. The review will focus on various types of oxidative stress-mediated kidney diseases, in which increased ROS production and/or
decreased antioxidants can be induced some kidney disease.

It can be seen from the attached manuscript that the author accepted all the suggestions given by other reviewers. That is why the manuscript is now more complete and very "fluent" to read.

Regardless, I also give my opinion on this review work.

General comments:

Title

The title is appropriate, precise and clear for readers.

Abstract

Written clearly and understandably. In brief, Includes all the elements for understanding what is written in the manuscript. Comments of other reviewers are included.

Manuscript

The author in the manuscript provides basic information on the metabolism of ceramides, metabolic pathways of sphingolipids and enzymes involved in these processes. For me, the Interaction between ROS, antioxidants and ceramide in oxidant-induced kidney injury was especially interesting. It is also interesting that decreased glutathione (GSH) and glutathione redox index (GSH/GSSG) enhanced nSMase activity.

Dealing with renal I/R we found that I/R, in most cases, caused an increase in FA concentration (C15:0, C16:1, C17:0, C18:1 c + t, C18:2 c + t, C22:0, and C22:6 + C22:1), with a decrease only in the case of C14:0 and C16:0, and no changes in the case of C18:0. It is known that I/R elevates ROS generation and decreases antioxidant defense through antioxidant enzyme gene downregulation. Studies have reported the loss of the MnSOD enzyme activity following I/R, with the MnSOD inactivation prior to the onset of renal damage, which suggests

an upstream of oxidative damage to renal damage.

We also investigated the protein expression of Nrf2, which has been shown to be the most
important inducible transcription factor that exerts protective effects against renal ischemic damage by regulating the endogenous antioxidant system. Once activated in cytosol, Nrf2 trans-activates antioxidant response elements, which further promote the CuZnSOD, GSH-Px, GST, CAT, and HO-1 expression increase. Several lines of evidence indicate that induced expression of HO-1 attenuates cellular damage and reduces apoptosis, while HO-1 inhibition aggravates it. Our results showed that I/R decreased kidney Nrf2 (16%) and HO-1 (31%) expression levels by 16% and 31%, respectively.

Cellular mechanisms that result in apoptosis represent a cascade of numerous amplifying steps, including activation of pro-apoptotic Bax protein, a member of the Bcl2 protein family. It has been suggested that a delicate interplay between anti- and pro-apoptotic members of the Bcl2 family is necessary for the repair of the damaged renal cells after an ischemic insult. Moreover, it has been reported that the over expression of anti-apoptotic Bcl2 can block both apoptosis and necrosis, as well as protect ischemic tissue against I/R-induced oxidative stress. We also found that meldonium pretreatment reduced Bax/Bcl2 ratio increase for 35%.

I think it would be useful to note this in the context of a possible link between oxidative stress and ceramide metabolism in appropriate places in the text.

Suggestion

This is just one example of how antioxidants can affect the metabolism of ceramides and I think that it will be very usefully to cite following reference together with above mentioned explanations with explanation in places where the author thinks best.

Đurašević S., Stojković M., Bogdanović Lj., Pavlović S., Borković-Mitić S., Grigorov I., Bogojević D., Jasnić N., Tosti T., Đurović S., Đorđević J., Todorović Z. (2019). The effects of meldonium on the renal acute ischemia/reperfusion injury in rats. Int. J. Mol. Sci., 20, 5747; DOI:10.3390/ijms20225747

There are no specific comments in the text.

Conclusion of the Reviewer

The author accepted and in detail explained the suggestions of other reviewers, So, in my opinion, this is a very high quality review manuscript and I suggest to the editor to accept this MS [IJMS] Manuscript ID_ijms-1640020 for publication in this form, but with the proproposed suggestion.

General conclusion: Acceptable for publication.

Dear Editor,

The manuscript [IJMS] Manuscript ID_ijms-1640020entitled: “A Rheostat of Ceramide and Sphingosine-1-Phosphate as a 2 Determinant of Oxidative Stress-Mediated Kidney Injury” by author Norishi Ueda  represent review article that summarize the current data considering a role of interaction between ROS-antioxidants and ceramide-SphKs/S1P and of a ceramide-S1P rheostat in the regulation of oxidant-induced kidney injury. The review will focus on various types of oxidative stress-mediated kidney diseases, in which increased ROS production and/or
decreased antioxidants can be induced some kidney disease.

It can be seen from the attached manuscript that the author accepted all the suggestions given by other reviewers. That is why the manuscript is now more complete and very "fluent" to read.

Regardless, I also give my opinion on this review work.

General comments:

Title

The title is appropriate, precise and clear for readers.

Abstract

Written clearly and understandably. In brief, Includes all the elements for understanding what is written in the manuscript. Comments of other reviewers are included.

Manuscript

The author in the manuscript provides basic information on the metabolism of ceramides, metabolic pathways of sphingolipids and enzymes involved in these processes. For me, the Interaction between ROS, antioxidants and ceramide in oxidant-induced kidney injury was especially interesting. It is also interesting that decreased glutathione (GSH) and glutathione redox index (GSH/GSSG) enhanced nSMase activity.

Dealing with renal I/R we found that I/R, in most cases, caused an increase in FA concentration (C15:0, C16:1, C17:0, C18:1 c + t, C18:2 c + t, C22:0, and C22:6 + C22:1), with a decrease only in the case of C14:0 and C16:0, and no changes in the case of C18:0. It is known that I/R elevates ROS generation and decreases antioxidant defense through antioxidant enzyme gene downregulation. Studies have reported the loss of the MnSOD enzyme activity following I/R, with the MnSOD inactivation prior to the onset of renal damage, which suggests

an upstream of oxidative damage to renal damage.

We also investigated the protein expression of Nrf2, which has been shown to be the most
important inducible transcription factor that exerts protective effects against renal ischemic damage by regulating the endogenous antioxidant system. Once activated in cytosol, Nrf2 trans-activates antioxidant response elements, which further promote the CuZnSOD, GSH-Px, GST, CAT, and HO-1 expression increase. Several lines of evidence indicate that induced expression of HO-1 attenuates cellular damage and reduces apoptosis, while HO-1 inhibition aggravates it. Our results showed that I/R decreased kidney Nrf2 (16%) and HO-1 (31%) expression levels by 16% and 31%, respectively.

Cellular mechanisms that result in apoptosis represent a cascade of numerous amplifying steps, including activation of pro-apoptotic Bax protein, a member of the Bcl2 protein family. It has been suggested that a delicate interplay between anti- and pro-apoptotic members of the Bcl2 family is necessary for the repair of the damaged renal cells after an ischemic insult. Moreover, it has been reported that the over expression of anti-apoptotic Bcl2 can block both apoptosis and necrosis, as well as protect ischemic tissue against I/R-induced oxidative stress. We also found that meldonium pretreatment reduced Bax/Bcl2 ratio increase for 35%.

I think it would be useful to note this in the context of a possible link between oxidative stress and ceramide metabolism in appropriate places in the text.

Suggestion

This is just one example of how antioxidants can affect the metabolism of ceramides and I think that it will be very usefully to cite following reference together with above mentioned explanations with explanation in places where the author thinks best.

Đurašević S., Stojković M., Bogdanović Lj., Pavlović S., Borković-Mitić S., Grigorov I., Bogojević D., Jasnić N., Tosti T., Đurović S., Đorđević J., Todorović Z. (2019). The effects of meldonium on the renal acute ischemia/reperfusion injury in rats. Int. J. Mol. Sci., 20, 5747; DOI:10.3390/ijms20225747

There are no specific comments in the text.

Conclusion of the Reviewer

The author accepted and in detail explained the suggestions of other reviewers, So, in my opinion, this is a very high quality review manuscript and I suggest to the editor to accept this MS [IJMS] Manuscript ID_ijms-1640020 for publication in this form, but with the proproposed suggestion.

General conclusion: Acceptable for publication.

Dear Editor,

The manuscript [IJMS] Manuscript ID_ijms-1640020entitled: “A Rheostat of Ceramide and Sphingosine-1-Phosphate as a 2 Determinant of Oxidative Stress-Mediated Kidney Injury” by author Norishi Ueda  represent review article that summarize the current data considering a role of interaction between ROS-antioxidants and ceramide-SphKs/S1P and of a ceramide-S1P rheostat in the regulation of oxidant-induced kidney injury. The review will focus on various types of oxidative stress-mediated kidney diseases, in which increased ROS production and/or
decreased antioxidants can be induced some kidney disease.

It can be seen from the attached manuscript that the author accepted all the suggestions given by other reviewers. That is why the manuscript is now more complete and very "fluent" to read.

Regardless, I also give my opinion on this review work.

General comments:

Title

The title is appropriate, precise and clear for readers.

Abstract

Written clearly and understandably. In brief, Includes all the elements for understanding what is written in the manuscript. Comments of other reviewers are included.

Manuscript

The author in the manuscript provides basic information on the metabolism of ceramides, metabolic pathways of sphingolipids and enzymes involved in these processes. For me, the Interaction between ROS, antioxidants and ceramide in oxidant-induced kidney injury was especially interesting. It is also interesting that decreased glutathione (GSH) and glutathione redox index (GSH/GSSG) enhanced nSMase activity.

Dealing with renal I/R we found that I/R, in most cases, caused an increase in FA concentration (C15:0, C16:1, C17:0, C18:1 c + t, C18:2 c + t, C22:0, and C22:6 + C22:1), with a decrease only in the case of C14:0 and C16:0, and no changes in the case of C18:0. It is known that I/R elevates ROS generation and decreases antioxidant defense through antioxidant enzyme gene downregulation. Studies have reported the loss of the MnSOD enzyme activity following I/R, with the MnSOD inactivation prior to the onset of renal damage, which suggests

an upstream of oxidative damage to renal damage.

We also investigated the protein expression of Nrf2, which has been shown to be the most
important inducible transcription factor that exerts protective effects against renal ischemic damage by regulating the endogenous antioxidant system. Once activated in cytosol, Nrf2 trans-activates antioxidant response elements, which further promote the CuZnSOD, GSH-Px, GST, CAT, and HO-1 expression increase. Several lines of evidence indicate that induced expression of HO-1 attenuates cellular damage and reduces apoptosis, while HO-1 inhibition aggravates it. Our results showed that I/R decreased kidney Nrf2 (16%) and HO-1 (31%) expression levels by 16% and 31%, respectively.

Cellular mechanisms that result in apoptosis represent a cascade of numerous amplifying steps, including activation of pro-apoptotic Bax protein, a member of the Bcl2 protein family. It has been suggested that a delicate interplay between anti- and pro-apoptotic members of the Bcl2 family is necessary for the repair of the damaged renal cells after an ischemic insult. Moreover, it has been reported that the over expression of anti-apoptotic Bcl2 can block both apoptosis and necrosis, as well as protect ischemic tissue against I/R-induced oxidative stress. We also found that meldonium pretreatment reduced Bax/Bcl2 ratio increase for 35%.

I think it would be useful to note this in the context of a possible link between oxidative stress and ceramide metabolism in appropriate places in the text.

Suggestion

This is just one example of how antioxidants can affect the metabolism of ceramides and I think that it will be very usefully to cite following reference together with above mentioned explanations with explanation in places where the author thinks best.

Đurašević S., Stojković M., Bogdanović Lj., Pavlović S., Borković-Mitić S., Grigorov I., Bogojević D., Jasnić N., Tosti T., Đurović S., Đorđević J., Todorović Z. (2019). The effects of meldonium on the renal acute ischemia/reperfusion injury in rats. Int. J. Mol. Sci., 20, 5747; DOI:10.3390/ijms20225747

There are no specific comments in the text.

Conclusion of the Reviewer

The author accepted and in detail explained the suggestions of other reviewers, So, in my opinion, this is a very high quality review manuscript and I suggest to the editor to accept this MS [IJMS] Manuscript ID_ijms-1640020 for publication in this form, but with the proproposed suggestion.

General conclusion: Acceptable for publication.

Dear Editor,

The manuscript [IJMS] Manuscript ID_ijms-1640020entitled: “A Rheostat of Ceramide and Sphingosine-1-Phosphate as a 2 Determinant of Oxidative Stress-Mediated Kidney Injury” by author Norishi Ueda  represent review article that summarize the current data considering a role of interaction between ROS-antioxidants and ceramide-SphKs/S1P and of a ceramide-S1P rheostat in the regulation of oxidant-induced kidney injury. The review will focus on various types of oxidative stress-mediated kidney diseases, in which increased ROS production and/or
decreased antioxidants can be induced some kidney disease.

It can be seen from the attached manuscript that the author accepted all the suggestions given by other reviewers. That is why the manuscript is now more complete and very "fluent" to read.

Regardless, I also give my opinion on this review work.

General comments:

Title

The title is appropriate, precise and clear for readers.

Abstract

Written clearly and understandably. In brief, Includes all the elements for understanding what is written in the manuscript. Comments of other reviewers are included.

Manuscript

The author in the manuscript provides basic information on the metabolism of ceramides, metabolic pathways of sphingolipids and enzymes involved in these processes. For me, the Interaction between ROS, antioxidants and ceramide in oxidant-induced kidney injury was especially interesting. It is also interesting that decreased glutathione (GSH) and glutathione redox index (GSH/GSSG) enhanced nSMase activity.

Dealing with renal I/R we found that I/R, in most cases, caused an increase in FA concentration (C15:0, C16:1, C17:0, C18:1 c + t, C18:2 c + t, C22:0, and C22:6 + C22:1), with a decrease only in the case of C14:0 and C16:0, and no changes in the case of C18:0. It is known that I/R elevates ROS generation and decreases antioxidant defense through antioxidant enzyme gene downregulation. Studies have reported the loss of the MnSOD enzyme activity following I/R, with the MnSOD inactivation prior to the onset of renal damage, which suggests

an upstream of oxidative damage to renal damage.

We also investigated the protein expression of Nrf2, which has been shown to be the most
important inducible transcription factor that exerts protective effects against renal ischemic damage by regulating the endogenous antioxidant system. Once activated in cytosol, Nrf2 trans-activates antioxidant response elements, which further promote the CuZnSOD, GSH-Px, GST, CAT, and HO-1 expression increase. Several lines of evidence indicate that induced expression of HO-1 attenuates cellular damage and reduces apoptosis, while HO-1 inhibition aggravates it. Our results showed that I/R decreased kidney Nrf2 (16%) and HO-1 (31%) expression levels by 16% and 31%, respectively.

Cellular mechanisms that result in apoptosis represent a cascade of numerous amplifying steps, including activation of pro-apoptotic Bax protein, a member of the Bcl2 protein family. It has been suggested that a delicate interplay between anti- and pro-apoptotic members of the Bcl2 family is necessary for the repair of the damaged renal cells after an ischemic insult. Moreover, it has been reported that the over expression of anti-apoptotic Bcl2 can block both apoptosis and necrosis, as well as protect ischemic tissue against I/R-induced oxidative stress. We also found that meldonium pretreatment reduced Bax/Bcl2 ratio increase for 35%.

I think it would be useful to note this in the context of a possible link between oxidative stress and ceramide metabolism in appropriate places in the text.

Suggestion

This is just one example of how antioxidants can affect the metabolism of ceramides and I think that it will be very usefully to cite following reference together with above mentioned explanations with explanation in places where the author thinks best.

Đurašević S., Stojković M., Bogdanović Lj., Pavlović S., Borković-Mitić S., Grigorov I., Bogojević D., Jasnić N., Tosti T., Đurović S., Đorđević J., Todorović Z. (2019). The effects of meldonium on the renal acute ischemia/reperfusion injury in rats. Int. J. Mol. Sci., 20, 5747; DOI:10.3390/ijms20225747

There are no specific comments in the text.

Conclusion of the Reviewer

The author accepted and in detail explained the suggestions of other reviewers, So, in my opinion, this is a very high quality review manuscript and I suggest to the editor to accept this MS [IJMS] Manuscript ID_ijms-1640020 for publication in this form, but with the proproposed suggestion.

General conclusion: Acceptable for publication.

Author Response

Please note that all revised sentences are underlined and highlighted with yellow color background in the revised manuscript.

  1. The reviewer recommended that it will be very usefully to cite following reference: Đurašević et al. The effects of meldonium on the renal acute ischemia/reperfusion injury in rats. Int. J. Mol. Sci. 2019, 5747.

According to the suggestion, I cited this reference [Đurašević et al,  Int. J. Mol. Sci., 2019, 5747, ref#121] in the revised manuscript. I revised the manuscript as follows:

On page 9, line 304-

Nuclear factor-erythroid 2-related factor-2 (Nfr2) is the most important inducible transcription factor that exerts protective effects against oxidant-induced kidney injury by stimulating the endogenous antioxidants. Once activated in cytosol, Nrf2 transactivated antioxidant response elements, which further enhanced the expression and activity of antioxidants such as SOD, GSH, glutathione peroxidase (GPx), GST, catalase and heme oxygenase (HO)-1 [121].  I/R decreased the expression of Nrf2 and HO-1 in the kidney, leading to apoptosis/necrosis.  Meldonium, an anti-ischemic drug clinically used to treat myocardial and cerebral ischemia that shifts energy production from fatty acid oxidation to less oxygen-consuming glycolysis, increased the expression of Nrf2 and HO-1, thereby increasing the expression and activity of antioxidants such as SOD, GPx and GST, leading to protection against I/R-induced kidney injury [121]. Since meldonium inhibited I/R-induced renal formation of fatty acids [121] which are a basic component of all lipids including palmitate and SLs (e.g., ceramide), it would be interesting to determine whether this agent suppresses ceramide levels by inhibiting supply of fatty acids which is needed for ceramide synthesis [122]. In addition, it would be interesting to find out whether meldonium inhibits ceramide synthesis via modulation of the enzymes involved in ceramide metabolism induced by antioxidants through Nrf2 and HO-1 in oxidant-induced kidney injury. Further studies are needed to determine whether antioxidants and antioxidant agents that activate Nfr2 modulate enzymes involved in ceramide metabolism directly or through inhibition of ROS-induced signaling pathways.

  1. b) “We also found that meldonium pretreatment reduced Bax/Bcl2 ratio increase for 35%.”

The idea of this study has been cited in the revised manuscript as follows.

On page 11, line 418

In addition, meldonium that functions as an antioxidant decreased the ratio of Bax/Bcl-2, ameliorating I/R-induced kidney injury [121].

This manuscript is a resubmission of an earlier submission. The following is a list of the peer review reports and author responses from that submission.

Round 1

Reviewer 1 Report

I appreciate the effort in summarizing a great size of references in the fields. 

I have a few general comments on the structure of the review. I wish the author could address those properly, before publication.

1. Throughout the content, I believe that the section number 9, starting from line 799, is the only content that fits with the title. The rest of the content is a vast summary of research on ceramide/S1P in renal cells, which would not help readers understand the point of the review. Please consider elaborating on section 9, and trim down or remove other parts. 

2. In most paraphrasing or summarizing articles, the sentences were too determinant that many results were oversimplified. For instance, in most research, the lipid content (ceramides or sphingosine-1-phosphate) is simply associated with the pathology, without a clear mechanistic casualty relationship. As providing lipids or ROS causes pleiotropic effects and lipids/ROS changes can occur via numerous pathways, it is very difficult to directly interpret the relationship between the phenotype and specific lipid content. However, the author keeps interpreting the phenotype/pathology as a direct effect of lipid content changes. 

3. In many paraphrasing, the results are not tightly organized for the paragraphs. Thus it is very difficult to follow the points. It may happen because there are too many results summarized with too short (or no) introductions for each research. Please consider trimming down the references to the key results only. 

Reviewer 2 Report

In this review (ijms-1381575), the author made substantial research efforts to survey the literature and summarize the latest updates on the role of ROS-antioxidants in modulating the ceramide-sphingosine kinases/S1P rheostat and their role in the regulation of oxidative stress-mediated kidney disease. Moreover, a detailed classification and explanation of the enzymes that generate ceramide and S1P in the kidney and their function and relationships to the oxidant-induced kidney injury have been well described. The author emphasized and illustrated the different types of oxidative stress-mediated kidney diseases in which increased ROS production and/or decreased antioxidants. In addition, existing data on role of mitochondrial function in the regulation of ceramide and ROS formation and their roles in oxidant-induced kidney injury were compared and discussed. Overall, the surveyed literature and protocol applied to achieve its purpose are adequate, well-structured, well-presented, and actual. The results provided by the manuscript could be a real gain for researchers interested in developing new possible therapeutic strategies and drugs to combat oxidative stress-mediated kidney diseases. Considering the potential impact of the manuscript outcomes in the research world, I recommend that the manuscript be accepted for publication in Int. J. Mol. Sci. journal following the following revision and suggestions.

  • in line 56 the author should refer to the preferred or suggested studies in the literature for the details of oxidant status in oxidative stress-mediated kidney diseases.
  • the description of sphingolipid metabolism is not well presented. The authors have not cited any relevant studies. Most importantly, as I see, the discussion of the SL metabolism should include the most recent and advanced findings, e.g.,

Reviewer 3 Report

The review concerns the role of the metabolism of sphingolipids, especially ceramide and its phosphorylated S1P derivative, in kidney damage. The author draws attention to the role of reactive oxygen species in these phenomena.

It is an encyclopedic compendium of data on the above-mentioned processes. In in the first chapter, the Author gives a nice introduction to the metabolism of sphingolipids. Next he enumerates enzymes involved in these metabolism pathways and indicates their localization in kidney. This part is accompanied by Table 1 which repeats in clear way the data from this chapter. The main part of the review considers the ROS-ceramide-S1P interrelations and consist of chapters on interaction between ROS, antioxidants and ceramide in oxidant-induced kidney injury, role of mitochondria function in the regulation of ceramide and ROS formation, and in ceramide-induced apoptosis. Next, the interaction between ROS and S1P in oxidant-induced kidney injury is shown, followed by the role of mitochondria and cell signaling pathways in S1P-induced cellular function in injured kidneys. Finally, the rheostat of ceramide/S1P in the regulation of oxidative stress-mediated kidney injury is indicated. Each of this chapters consists of several sub-elements.

This statistics clearly shows, on one hand how broad is the scope of the review, and on the other hand, what details the Author tries to take into account. The amount of work done could be admired. Tables that clearly present the data described in the text and are a great advantage of the manuscript.

However, the review poses a challenge to the reader. Each chapter contains mainly lists of papers related to the a given topic. Unfortunately, there is no attempt to guide the reader through these data, nor is there any attempt (or minimal) to synthesize the data. What's more, the enormity of individual data translates into extremely complex Figures which do not facilitate the reception of work. Scheme no. 3 is bizarre in this respect.

In my opinion this review is not suitable for publishing and requires a really thorough amendments. The review constitutes a collection of data, but the Author should facilitate understanding of their meaning, ending with an attempt to obtain a more general picture of the discussed process. Above all, there is no need to repeat the data in Tables in the text. The Author  also should think over what he wants to show in Figures. These schemes should be informative but clear and emphasize main discoveries. For more complex issues, the Author can divide Figures into several panels showing individual pathways, to obtain a complex but transparent picture of the processes.

 There are also other minor concerns:

1. The overabundance of abbreviations is overwhelming, I recommend to make a list of them which should facilitate reading of the paper. This is a typical sentence form the manuscript:

“However, ceramide levels increased in diabetic  kidney, RTCs [100] and podocytes via SPT [101] and ECs via aSMase [102,103] and CerS 180 [104]”.

2. Sphingomyelin hydrolysis to ceramide by aSMAse and resynthesis of sphingomyelin by SMS2 in the plasma membrane should be mentioned in the overview the of sphingolipid metabolism

3. There are some shortcuts and errors:

Line 9: ROS produce ceramide … and inhibit S1P. Does it mean that ROS induce an increase of the ceramide level and decrease of the S1P level? Or do they induce activation and inhibition of respective enzymes?

Line 13: Ceramide …. increases ceramide- and voltage-dependent anion channels (activity of these channel?)

Line 48: Limited information is available of a role of interaction

Line: 88-89 (Fig. 1 legend) Sph…. is degraded by phosphorylation

Line 124, Table 2 title and the Table: Alteration of ceramide in oxidant-induced kidney injury. Does it mean alterations of the ceramide level?

Site for Cer alteration - ?

 Line 164: ACDase-knockout mice increased glomerular ceramide level.